# Comparison of Quality Changes in Eurasian Perch (*Perca fluviatilis* L.) Fillets Originated from Two Different Rearing Systems during Frozen and Refrigerated Storage

**DOI:** 10.3390/foods10061405

**Published:** 2021-06-17

**Authors:** Nima Hematyar, Jan Mraz, Vlastimil Stejskal, Sabine Sampels, Zuzana Linhartová, Marketa Prokesova, Frantisek Vacha, Martin Krizek, Eva Dadakova, Hanne Søndergård Møller, Trine Kastrup Dalsgaard

**Affiliations:** 1South Bohemian Research Center of Aquaculture and Biodiversity of Hydrocenoses, Faculty of Fisheries and Protection of Waters, Research Institute of Fish Culture and Hydrobiology, University of South Bohemia in Ceske Budejovice, Zátiší 728/II, 389 25 Vodňany, Czech Republic; 2South Bohemian Research Center of Aquaculture and Biodiversity of Hydrocenoses, Faculty of Fisheries and Protection of Waters, Institute of Aquaculture, University of South Bohemia in České Budějovice, Husova tř. 458/102, 370 05 České Budějovice, Czech Republic; jmraz@frov.jcu.cz (J.M.); stejskal@frov.jcu.cz (V.S.); linhartova@frov.jcu.cz (Z.L.); mprokesova@frov.jcu.cz (M.P.); 3Department of Molecular Sciences, Swedish University of Agricultural Sciences, P.O. Box 7015, 75007 Uppsala, Sweden; sabine.sampels@slu.se; 4Faculty of Agriculture, University of South Bohemia in Ceske Budejovice, Studentská 1668, 370 05 České Budějovice, Czech Republic; vachaf@psp.cz (F.V.); krizek@zf.jcu.cz (M.K.); dadakova@zf.jcu.cz (E.D.); 5Department of Food Science, Aarhus University, Agro Food Park 48, 8200 Aarhus N, Denmark; hsm@food.au.dk (H.S.M.); trine.dalsgaard@food.au.dk (T.K.D.); 6CiFood—Centre for Innovative Food Research, Aarhus University, Agro Food Park 48, 8200 Aarhus N, Denmark; 7CBIO—Centre for Circular Bioeconomy, Blichers Allé, 8830 Tjele, Denmark

**Keywords:** rearing systems, fillet quality, protein characterization, hardness, liquid loss

## Abstract

The current knowledge on how different Eurasian perch rearing systems impact the final fillet quality is scant. Therefore, two domestic storage conditions were investigated—10 months frozen (−20 °C) and 12 days refrigerated (+4 °C) storage conditions—in order to determine (i) how the choice of rearing system affects fillets quality during different processing conditions and (ii) if oxidative changes and other quality parameters were interactive. For the proposed idea, proteome analysis, oxidative changes, and some quality parameters were considered in this study. Sodium dodecyl-sulfate polyacrylamide gel electrophoresis (SDS-PAGE) indicated a higher loss of protein in the frozen fillets from ponds (PF) than the fillets from recirculating aquaculture systems (RAS) (RF). Western blot showed a higher protein carbonyls level in RF compared to PF, which was confirmed by the total protein carbonyls during frozen storage. PF indicated less liquid loss, hardness, and oxidation progress than RF in both storage conditions. The biogenic amines index (BAI) in the fillets from either origin showed acceptable levels during storage at +4 °C. Furthermore, the n-3/n-6 ratio was similar for both fillets. The deterioration of fillets during frozen storage was mainly caused by formation of ice crystals followed by protein oxidation, while protein oxidation was the main concern during refrigerated storage confirmed by principal component analysis (PCA) analysis.

## 1. Introduction

The production of Eurasian perch (*Perca fluviatilis* L.) in aquaculture was 585 tons in 2016 [1] (https://www.fishbase.in/search.php, accessed on 12 June 2021). Due to the high market price of Eurasian perch (*Perca fluviatilis* L.) fillet, it is considered a valuable commercial fish species [2].

Through the last two decades, recirculating aquaculture systems (RAS) have been expanded for different of European inland aquaculture [3]. In this rearing system, stable temperature (23 °C), uniform photoperiod (12L:12D), high stocking density (up to 60 kg m-3), and commercially formulated feeds are used [4]. Perch reared in RAS has a better growth rate as a result of optimal feed, optimal temperature, and optimal oxygen levels or suppressed maturity with a constant thermal and light regime [5]. On the other hand, fish reared in ponds with fluctuating temperatures have different growth rates and they grow on more natural feed like zooplankton and chironomids, which are responsible for different colors in fish fillets [6].

Furthermore, the type of rearing system might have a direct or indirect impact on fish flesh quality factors, such as fiber development, color, and textural parameters. The impact of rearing systems on the textural and nutritional quality of pike perch and perch fillets has been reported before [7,8]. Therefore, considering the effect of rearing systems on fillet quality gives a good perspective about optimizing the conditions for fish farming.

Beside the rearing system, several factors might affect the fish fillet quality such as the slaughtering and packaging method as well as storage conditions [9,10]. Freezing is used as a common method in order to keep the nutritional and sensory quality of fish flesh for a long storage time. Large ice crystal that are distributed irregularly due to the improper freezing would be the main concern of fillet deterioration while, small and regularly distributed ice crystal by quick freezing can prevent unpleasant fillet quality as well as protein structure changes [11]. Additionally, refrigerated storage is widely used before technological processes or consumption for the protection of fish quality. Even though during frozen and refrigerated storage the growth of spoilage microorganisms, the enzymatic activity, and the chemical reactions are slower, over time they still lead to diminished fish quality.

It has been proposed that fish fillet softening occurs due to three mechanisms, namely, ice crystal formation during frozen storage [12], endogenous enzyme activity that leads to protein degradation during refrigerated storage [13], and the progress of lipid and protein oxidation and the reaction product of these in both mentioned conditions [14]. Rancidity of lipids leads to a decrease in the acceptability of fish fillets and makes fillets prone to oxidation, especially the ones containing a high amount of polyunsaturated fatty acids (PUFA). Moreover, it is assumed that actin and myosin are the main proteins responsible for the functional properties of muscle foods, and during frozen and refrigerated storage, both proteins can undergo an aggregation reaction and decline in water-holding capacity (WHC) and hardness in the muscle [15,16]. [17] showed the loss of myosin light chain (MLC) and α-actin bands with sodium dodecyl-sulfate polyacrylamide gel electrophoresis (SDS-PAGE) in the carp (*Labeo rohita*) fillet during 6 months of storage at −20 °C.

On the other hand, a review of the available literature revealed a lack of comprehensive research on biogenic amines formation in perch fillet. There are two main reasons to determine biogenic amines in fish: (1) their potential toxicity and (2) their use as food quality indicators [18]. Biogenic amines (BAs) are formed by the decomposition of the protein matrix, and their content can be used as an additional criterion for the microbial degradation of the flesh. In this respect, especially diamines such as putrescine (PUT), cadaverine (CAD), and tyramine (TYM) are important. [19] reported that the PUT and CAD contents are good freshness quality indicators in perch and trout, since both amines were correlated with sensory evaluation levels. Additionally, histamine (HIM) and TYM formation indicates improper storage temperature. The limitation level of total biogenic amines in fish is 300 mg/kg and more is considered detrimental to human health [20]. HIM has been widely studied in the literature for similar reasons, but most importantly, for its toxicity [21]. The biogenic amines index (BAI) is used as an indicator of freshness. In the case of fish fillets, 0 and 1 reflect freshness, a score between 1 and 10 is acceptable, and more than 10 demonstrates decomposition of the fish fillet [22].

The present study was designed to determine the quality of Eurasian perch fillet, which was reared in two different rearing systems during frozen and refrigerated storage conditions. At present, little is known about the role of specified proteins on fillet quality affected by different rearing systems. Furthermore, we designed a way to monitor the modification of specified proteins influenced by rearing systems as well as storage conditions. The hypotheses were as follows: (i) that the rearing system would affect the overall quality of fish flesh upon post-mortem storage and (ii) oxidative changes and other quality parameters were interactive. The consequent repercussions of the mentioned modifications on the fillet quality were assessed with respect to a proteomic approach together with the characterization of protein and lipid oxidation, hardness, WHC, rigor index, pH, and biogenic amines content. Furthermore, the relationship between lipid and protein oxidation products was considered.

## 2. Materials and Methods

### 2.1. Experimental Design

In the present study, 88 Eurasian perch from two different rearing systems (traditional pond culture and RAS; average 183 g and 211 g for 30 fish in weight and average 23.46 and 24.37 cm in length for RAS and pond fish, respectively), with the pond age 2+ and RAS age 1+, were used. 

Pond-reared perch were obtained from the fish farm Rybarstvi Nove Hrady Inc. (pond Zar) located in the south of the Czech Republic (48°8′ N, 14°7′ E). Fish were reared in polyculture (natural production of 240 kg ha-1) with common carp (*Cyprinus carpio*) as a main fish species. The feed source was natural prey with the main forage fish being topmouth gudgeon (*Pseudorasbora parva*), roach (*Rutilus rutilus*), and benthic organisms. The fish were harvested in mid-October during the common Czech harvest period. 

The fish from RAS were reared at the University of South Bohemia, Facility of Fisheries and Protection of Waters, České Budějovice, Czech Republic (USB, FFPW) and fed commercial feed (Inicio plus, BioMar, Nersac, France) with the following nutrient values: protein 52%, fat 23%, fiber 0.9%, carbohydrates (NFE) 12%, gross energy 23.5 MJ kg-1, and digestible energy 20.6 MJ kg-1.

#### 2.1.1. Sample Preparation for Textural and Chemical Analysis

Fish from both rearing systems were kept 7 days in flow-through tanks without feeding before slaughtering. Fish (both from RAS and pond) were kept at the faculty in tanks at the same temperature (20 °C) for one week before being slaughtered (fish were killed by blows to the head, bled and gutted, skinned, washed subsequently, and filleted on the faculty premises by one trained person. Fish fillets (*n* = 6) were randomly packed in plastic bags, labelled, and stored in a normal freezer at (−20 °C) for 10 months and a refrigerator at +4 °C for 12 days. Six, randomly chosen right fillets from each group were used for hardness, liquid loss, and color analyses throughout the experiment, and chemical analyses were done on left fillets. Textural, chemical, and proteomics analyses were executed on the first day and after 4, 8, and 10 months and 0, 4, 8, and 12 days of frozen and refrigerated storage, respectively. For the textural analysis, we considered fresh fillet (immediately after slaughtering) as time 0. In case of frozen samples, all raw fillets were defrosted (kept at +4 °C overnight) in advance, and the texture was measured. Regarding the chemical analysis, 6 whole fillets of frozen fillets from each time point were defrosted (kept at +4 °C overnight) and minced separately to obtain homogeneous and representative samples that were stored at −80 °C until further analysis.

#### 2.1.2. Sample Preparation for Rigor Index

After stunning, randomly, 10 fish from each rearing system (RAS and pond) were separated and kept on ice in the fridge, and then the rigor index was monitored at the specified interval times (0, 6, 9, 12, 24, 36,48, 60, 72, 96, 120, and 142 h). 

### 2.2. Fatty Acid Profile

Lipid extraction was performed according to [23], with a slight modification. The samples were semi-thawed, and sub-samples of approximately 1 g of fish fillet were sampled for the extraction. The samples were homogenized for 3 × 30 s in 10 mL of hexane: isopropanol (3:2 *v/v*) using an Ultra Turrax (T25; IKA-Werke Janke & Kunkel GmbH & Co., Staufen, Germany), and 6.5 mL Na2SO4 solution (0.47 M) was added. The homogenate was left to separate at 4 °C for 20 min and the upper phase was then transferred to a new tube and evaporated under N2. The lipid content of the samples was determined gravimetrically from this total extracted lipid, which was subsequently dissolved in 1 mL of hexane. Fatty acids (FAs) from the total lipids were methylated with a boron trifluoride-methanol complex (BF3) [24]. To each sample, 2 mL of a 0.01 M solution of NaOH in dry methanol was added, and the samples were heated for 10 min at 60 °C. Subsequently, 3 mL of BF3 reagent was added, and the samples were reheated at 60 °C for 10 min. Thereafter, the tubes were cooled in ice water and 2 mL of a 3.42 M NaCl solution in water was added to all tubes. The FA methyl esters (FAMEs) were extracted with 2 mL of hexane and the upper layer was transferred to a new tube and evaporated under nitrogen to dryness. The lipids were dissolved in 0.5 mL of hexane and stored at 80 °C until gas chromatographic (GC) analysis. The FAMEs were then analyzed using a gas chromatograph (Trace Ultra FID; Thermo Scientific, Milan, Italy) equipped with a flame ionization detector and a PVT injector, using a BPX 70 column (SGE Inc., Austin, TX, USA), with a length of 50 m, i.d. of 0.22 mm, and film thickness of 0.25 µm [25]. The GC was programmed with a constant gas flow of 1.2 mL/min and a temperature program that started at 70 °C for 0.5 min, followed by a ramp of 30 °C/min up to 150 °C, a second ramp with a rate of 2 °C/min up to 220 °C, and a final constant time of 11 min at 220 °C. The injector and detector temperatures were set at 150 °C and 250 °C, respectively. The injector was set in the splitless mode, with a splitless time of 0.8 min and a split flow of 25 mL/minute. The peaks were identified by comparing their retention times with those of the standard mixture GLC-68D (Nu-Chek Prep, Elysian, MN, USA) and other authentic standards (Nu-Chek Prep, Elysian, MN, USA; Larodan, Sweden).

#### Calculations

The lipid quality indexes related to ischemic heart disease risk, including the thrombogenicity index (TI) and atherogenic index (AI), were calculated according to [26] using Equations (1) and (2):AI = [12:0 + (4 × 14:0) + 16:0]/[(PUFAn-6 + *n* 3) + 18:1 + other MUFA](1)
TI = [14:0 + 16:0 + 18:0]/[0.5 × 18:1 + 0.5 × other MUFA + 0.5 × *n*-6PUFA + 3 × *n*-3PUFA + (*n*-3PUFA/6PUFA)](2)

### 2.3. Rigor Index

The rigor mortis development was measured by Cuttinger’s method (tail drop) [27] on 10 fish from each rearing system. The fillets were kept on ice and in the fridge during the rigor index analyses. The rigor index (Ir) was calculated by the formula Ir = [(Lo-Lt)/Lo] × 100, where L represents the vertical drop (cm) of the tail when half of the fish fork length is placed on the edge of a table as a function of time. The tail drop at the beginning of the experiment is Lo, while Lt represents measurements throughout the experiment (t = 0–142 h). T= 0 means immediately after slaughtering.

### 2.4. pH

The pH of fish fillets (*n* = 6) was measured by inserting a pH probe (Testo 206, Lenzkirch, Germany) into the upper mass of the fillet, just behind the head

### 2.5. Biogenic Amines

Samples (5 fillets from each rearing system) were homogenized using an Ultra-Turrax T25 homogenizer (Ika Labortechnik, Staufen, Germany). Biogenic amines were extracted from the homogenized material with diluted perchloric acid, p.a. (0.6 M). After filtration, the volume was made up to 150 mL with perchloric acid. The amines were determined as dansyl derivatives after derivatization with dansyl chloride by UPLC. The procedure has been described in detail by [28]. The biogenic amines index (BAI) was calculated using Equation (3) (contents of BAs are given in mg kg^−1^):BAI = (mg kg^−1^ Histamine + mg kg^−1^ Putrescine + mg kg^−1^ Cadaverine)/(1 + mg kg^−1^ Spermine + mg kg^−1^ Spermidine)(3)

### 2.6. Hardness Analysis

Hardness analysis (*n* = 6 fillets from each rearing system) was performed instrumentally using a texture analyzer (TA-XT. Plus, Stable Micro systems UK) by pressing a flat-ended cylinder (10 mm diameter, type P/10) into the section of fillet below the dorsal fin perpendicular to the muscle fibers at a speed of 2 mm/s until the fillet was compressed to 50% of its original thickness. Hardness was defined as the maximum force detected during first compression, expressed in grams.

### 2.7. Liquid Loss

Liquid loss was measured on *n* = 6 fillets from each rearing system with a texture analyzer (TA-XT. Plus, Stable Micro systems UK) by pressing a flat-ended cylinder (10 mm diameter, type P/10) into the fillet below the dorsal fin perpendicular to the muscle fibers at a speed 2 mm/s until it reached 50% of the fillet height and held for 60 s. A dry, pre-weighed filter paper was placed under the sample. The filter paper was then weighed immediately after the test, with and without the fish piece, and the liquid loss was calculated.

### 2.8. Fillet Color

Fillet color was evaluated by the Minolta Colorimeter (Spectro Photo Meter, CM- 600d, Konica, Japan) as described by [29]. We used the dorsal part of the fillet (*n* = 6) from each rearing system for color analysis. The parameters L* (lightness), a* (red-green spectrum), and b* (yellow-blue spectrum) were used to study the color properties of the fillet surface, where L* is the luminance score and ranges from zero (black) to 100 (white), and a* and b* are chromatic scores.

### 2.9. Protein Carbonyls Quantification with 2,4-Dinitrophenylhydrazine

Carbonyl content was considered for protein oxidation following incubation with DNPH in 2 N hydrochloric acid with a slightly improved method described by [30]. Meat (0.5–1 g) was homogenized in 10 mL KCl (0.15 M) using an UltraTurrax (Janke & Kunkel, T25IKA-Labortechnik, Staufen, Germany,) for 3 × 20 s at a speed of approximately 20,000 g. Homogenate solution (100 µL) was pipetted into 2 mL centrifuge tubes (Eppendorf), and 1 mL trichloroacetic acid 10% (TCA) was added to each tube and then centrifuged at 5000 rpm for 5 min. The supernatant was removed. One mL HCl (2N) was added to the blank, and then 1mL DNPH in 2 M HC1 was added, and they were allowed to stand at room temperature for 1 h, with vortexing every 20 min. Then, 1 mL trichloroacetic acid (10%) was added and centrifuged in the tubes in a tabletop micro centrifuge (3000× *g*) for 5 min, and the supernatant was discarded. The pellets were centrifuged with 1 mL ethanol-ethyl acetate (1:2, *v/v*) in 10,000 rpm for 5 min at 4 °C. The pellets were washed 2 times with 1 mL ethanol-ethyl acetate (1:2, *v/v*) to remove free reagents. The precipitated protein was redissolved in 1.5 mL guanidine solution (6 M) and centrifuged in the micro centrifuge for 2 min at 3000 *g*. The spectrum was read against the complementary blank in the case of cruder samples or against water in the case of purified proteins. The carbonyl concentration was analyzed as DNPH, calculated based on the protein hydrazine absorption in 21.0 mM-1 cm-1 at 370 nm. The concentration of protein was calculated at 280 nm in the same sample, and then, bovine serum albumin was used to quantify it as a standard.

### 2.10. Thiobarbituric Acid Reactive Substances

Thiobarbituric acid-reactive substances (TBARS) were analyzed according to the method by [31]. The semi-frozen samples were chopped, the visible fat and connective tissues were separated, and then, for extraction, approximately 1 g of muscle tissue was taken. The samples were mixed with 9.1 mL (0.61 mol/L) of trichloroacetic acid (TCA) solution and 0.2 mL (0.09 mol/L) of butylated hydroxytoluene (BHT) in methanol, using an UltraTurrax (Janke & Kunkel, Staufen, Germany, T25IKA-Labortechnik,) for 3 × 20 s at a speed of 14,000 rpm. Subsequently, the mixed sample was filtered through a Munktell paper (Munktell Filter AB, Grycksbo, Sweden). Two times, 1.5 mL of the mixed and filtrated sample were transferred to new tubes, and 1.5 mL water was added to the first (sample blank), and 1.5 mL of thiobarbituric acid (TBA) solution (0.02 mol/L) was added to the second (test sample). The reaction was completed for 15–20 h (overnight) in darkness at room temperature (20 °C), the reaction complex was detected at a wavelength of 530 nm against the sample blank using a UV-visual spectrophotometer (Specord 210; Analytik Jena, Germany). The amount of TBARS was expressed as malondialdehyde (MDA) (µg/g) based on the external standard curve.

### 2.11. Extraction of Muscle Proteins

Approximately 100 mg of frozen fish muscle was cut and weighed at −20 °C to minimize the degradation of protein. The frozen muscle was mixed in 500 µL in PBS (saline solution with a phosphate buffer concentration of 0.01 M and a sodium chloride concentration of 0.154 M, pH = 7.4) and transferred to an Eppendorf tube.

### 2.12. SDS-PAGE

The sodium dodecyl sulfate polyacrylamide gel electrophoresis (SDS-PAGE) was performed according to the method of [32]. Each sample (20 µL) was homogenized with Laemmli sample buffer and used the standard curve to reach 2 µg/µL as the final protein concentration, followed by 2 min heating at 95 °C. Subsequently, the samples were loaded on to a 10% Criterion Tris glycine Gel (Bio-Rad, Hercules, CA, USA) at a constant electrical potential of 200 V. As marker proteins, the Spectra Multicolor Broad Range Protein Ladder (15–220 kDa) (Thermo Scientific, Rockford, IL, USA) was used. Subsequently, the electrophoresis gel was stained with 0.5% Coomassie Brilliant Blue G-250 (Bull Korean Chem Soc. 2002).

### 2.13. Native Polyacrylamide Gel Electrophoresis (PAGE)

Samples were homogenized with lysis buffer to 5 µg/µL as the final protein concentration. Then, they were incubated for 15 min on ice followed by centrifuge at 20,000× *g* for 30 min at room temperature. Samples (15 µL) for native PAGE analysis were used without heating treatment. The kit (Life Technologies) and the protein were separated on 3–12% Native PAGE gel (Life Technologies) using the Native PAGE running buffer kit (Life Technologies). Gels were stained with 0.5% Coomassie Brilliant Blue G-250 (Bull Korean Chem Soc. Seoul. 2002).

### 2.14. Reaction with 2,4-Dinitrophenylhydrazine and Immunoblotting

For immunoblotting, the 2,4-dinitrophenyl hydrazine (DNPH) reaction was applied directly on the protein mix or on the protein sarcoplasmic fraction (low salt-soluble protein) acquired by centrifugation of the homogenate protein for 3 min at 12,600× *g*. The supernatant was used for analysis.

Protein carbonyls were derivatized by combining 20 µL of the sample (1:2) with 10% TFA, 12% SDS, and 10 mM DNPH and incubated for 30 min at room temperature. By adding 40 µL of neutralization buffer (1:2) containing 30% Glycerol, 2 M Tris-base, and 20 mM DTE, the reaction was stopped before being separated on an SDS-PAGE.

Before the DNPH reaction in immunoblotting, the concentration of protein was adjusted to 10 mg/mL by using the BCA kit (Pierce, Rockford, IL, USA). The obtained samples were centrifuged at 12,600× g for 3 min and then loaded on the gel. The protein bands from the gel were transferred to a polyvinylidene difluoride (PVDF) membrane of 0.2 µm (Bio-Rad, Laboratories, CA, USA) using the Trans-Blot SD, semi-dry transfer cell, 0.35 A, at max 50 V for 60 min (Bio-Rad, Laboratories, USA). After transfer, the membranes were blocked in 5% skim milk in a Tris-buffered saline (TBS) buffer (0.137 M NaCl and 20 mM Tris-HCl, pH 8.0) and incubated with a 1:16000 dilution of rabbit anti-DNP (Sigma Aldrich, Taufkirchen, Germany), in TBS overnight at +4 °C. The membranes were washed in TBS and incubated in a 1:8000 dilution of the secondary antibody, which was peroxidase-conjugated swine anti-rabbit (DAKO Denmark A/S). After washing in TBS, the blot was developed using the ECL ± kit (Bio-Rad Laboratories, USA). Image analysis of gels and blots were performed using the software Image lab (Molecular Imager Chemi Doc XRS+, Bio-Rad Laboratories, Hercules, CA, USA).

### 2.15. Protein Identification by Matrix-Assisted Laser Desorption/Ionization-Time of Flight (MALDI-TOF) Analysis

Bands of interest were cut from the gels were transferred to in-gel trypsin digestion as described by [33]. Prior to analysis, the samples were desalted and concentrated using custom-made C18 columns and eluted directly on the target plate using 70% acetonitrile, 1% formic acid, and 10 g/l α-cyano-4-hydroxycinnamic acid (Sigma–Aldrich, St. Louise, MO, USA).

The peptides mixtures were analyzed by using MALDI-TOF tandem mass spectrometry (MS/MS) (MALDI-TOF MS-MS, Autoflex Speed, Bruker Daltonics, Bremen, Germany). For internal calibration a peptide standard mixture from 1000–3000 Da (Bruker Daltonics, Bremen, Germany). Protein identification was performed by using Peptide Mass Fingerprinting using the in-house Mascot server according to [34].

### 2.16. Statistical Analysis

Statistical evaluation of results was performed using two-way ANOVA analysis in the Statistica CZ v. 12 software package. Tukey’s HSD tests were applied to find significant differences between all data. The significance level was considered at *p* < 0.05 and all results were presented as mean ± S.D. In addition, a principal component analysis (PCA) using the Unscrambler × 10.1 (Camo Process A/S, Oslo, Norway) was performed for the samples were stored in the fridge, and frozen samples were stored separately. For samples from refrigerated storage rigor, liquid loss, hardness, pH, color values, TBARS, carbonyls, and biogenic amines were included. Values from all timepoints were used except for biogenic amines, where only values from days 7 and 14 were included as those correspond to the storage period. For the frozen samples, rigor, liquid loss, hardness, pH, color values, TBARS, carbonyls, MHC, nebulin, actin, and MLC from all timepoints were included, but MHC, nebulin, actin, and MLC were added as one replicate only. Values for MHC, nebulin, actin, MLC, hardness, L*, and rigor are weighted as 1/(stdev) in the respective analyses. Full cross validation was used as the validation model. The number of PCs finally used was determined to be 5.

## 3. Results

### 3.1. Fat Content and Fatty Acids Profile

The fat content in the fillets from the fish reared in RAS (RF) and ponds (PF) did not show statistically different values (*p* < 0.05; Table 1).

The percentages of saturated fatty acids (SFA), monounsaturated fatty acids (MUFA), and polyunsaturated fatty acids (PUFA) in the RF and PF constituted 25.85% and 30.85%, 43.3% and 25.72%, and 26.56% and 47.72%, respectively. The total amount of SFA was the same in both groups. Palmitic acid (16:0) constituted the majority of SFA. RF showed a significantly higher amount of myristic acid (14:0) and PF had a significantly higher amount of stearic acid (18:0). The prominent MUFA was oleic acid in the fillets from both systems. RF showed a significantly higher amount of Myristoleic acid (14:1), oleic acid (18:1n-9), as well as gondoic acid (20:1n-9) fatty acids, while PF indicated a significantly higher amount of palmitoleic acid (16:1) and vaccenic acid (18:1n-7). We observed a significant difference in the total amount of PUFA between fillets from the two systems. Docosahexaenoic (DHA; C:22:6n-3) and eicosapentaenoic (EPA; C:20:5n-3) were identified as predominant n-3 PUFAs in the fillets from both systems. A significantly higher amount of linoleic (LA, 18:2 n-6) and docosahexaenoic acid (DHA, 22:6 n-3) was observed in RF. However, PF revealed a significantly higher amount of alpha-linoleic acid (aLA, 18:3 n-3), docosapentaenoic acid (22:5 n-3), arachidonic acid (AA) (20:4 n-6), eicosatrienoic acid (20:3 n-3), and eicosapentaenoic acid (EPA,) (20:5 n-3). The amount of eicosadienoic acid (20:2 n-6) was the same in fillets from both systems.

In the current study, we could not observe any difference (*p* > 0.05) in the thrombogenicity index (TI), which had values of 0.39 ± 0.02 for fillets from both rearing systems. Additionally, no significant difference was observed in the atherogenic index (AI), which reached 0.22 ± 0.01 for both fillets.

### 3.2. Autolytic Changes during Refrigerated Storage

#### 3.2.1. Rigor Index

The rigor index results during 142 h at +4 °C are shown in Figure 1. The onset of rigor was started after 6 and 9 h, in RF and PF, respectively. Additionally, a significant difference was observed between the time to reach the maximum rigor index in both systems (*p* < 0.05). Full rigor in the RF and PF was observed after 12 and 24 h, respectively.

#### 3.2.2. pH

The results of pH in perch fillets from both rearing systems (RF and PF) are shown in (Figure 2). pH did not show any statistical differences during 140 h storage of RF and PF at +4 °C. Comparing each time point indicated a significantly (*p* < 0.05) higher pH in RF rather than PF from 72 to 140 h (Figure 2).

### 3.3. Biogenic Amines Contents

The concentrations of all amine contents in RF and PF during 28 days of refrigerated storage at +4 °C are presented in Table 2.

The content of PUT, CAD, TYM, and HIM increased with the elapsing time. On the other hand, in the initial stages of storage, there was a statistically significant difference between RF and PF. This is particularly evident during the first 14 days in the case of PUT and TYM changes. However, this trend was not observed in CAD. In RF, the PUT and TYM contents were always lower during the first 14 days compared to the PF, while after 14 days of storage, the differences in both types of fillets were not statistically significant, probably due to the more advanced degradation of the protein matrix. It is also important that HIM was not detected at +4 °C during the first 14 days of storage.

Regarding the biogenic amines index, we observed an increasing trend in fillets from both rearing systems during storage time. Additionally, PF showed a higher BAI compared to RF.

### 3.4. Hardness Changes during Frozen and Refrigerated Storage

Hardness did not show a significant difference between RF and PF; however, hardness declined significantly (*p* < 0.05) over time of frozen storage, while no changes were observed during refrigerated storage (Figure 3a,b).

The initial hardness values were 1533 (g) and 1239 (g), decreased to 790 and 518 during frozen storage, and declined to 1185 and 981 during refrigerated storage in the RF and PF, respectively. RF showed a higher hardness after 8 months pf frozen storage compared to PF.

A major reduction in hardness was observed after 4 months of frozen storage in both PF and RF (43% and 40%, respectively), where afterwards, the hardness changed less over time of storage.

### 3.5. Liquid Loss during Frozen and Refrigerated Storage

RF and PF indicated a significant (*p* < 0.05) increase of liquid loss during both storage conditions (Figure 4a,b).

The amount of liquid loss increased from 0.2% and 0.15% to 2.3% and 1.5% during frozen storage and enhanced to 0.67% and 0.56% during refrigerated time in the RF and the PF, respectively. In addition, a time-to-time comparison between RF and the RF did not show any significant differences until the 4th month of storage, but after the 8th and 10th month, we observed a considerable difference between the two rearing systems (*p* < 0.05; Figure 4a). During refrigerated storage at +4 °C, both RF and PF indicated a significant increase (*p* < 0.05) between day 0 and day 4 of storage (Figure 4b). For the rest of the time, liquid loss did not show any significant differences. Moreover, we did not observe any significant differences between RF and PF after 12 days. In summary, RF showed higher liquid loss compared to PF during both storage conditions.

### 3.6. Color Changes during Frozen and Refrigerated Storage

Table 3 and Table 4 show the color changes of the RF and PF during 10 months of frozen storage at −20 °C and post-mortem time (+4 °C). 

L* and b* values of the RF and PF enhanced significantly (*p* < 0.05) in the period of storage time. On the other hand, the a* value in both fillets decreased significantly during the whole period of storage, indicating a less reddish color.

During refrigerated storage, L* values for the PF decreased significantly from the first day until the 8th day (*p* < 0.05), while in RF, no statistical changes were found. In PF, the b* value was significantly declined on day 4 (*p* < 0.05), while values on day 8 and 12 were intermediate.

A comparison of the results at each time point between the RF and PF showed significant differences in all color values, which showed more lightness in the PF and more redness and yellowness in the RF.

### 3.7. Protein and Lipid Oxidation during Frozen and Refrigerated Storage

The total content of protein carbonyl increased significantly (*p* < 0.05) in the RF and PF during both storage conditions (frozen and refrigerated; Table 5 and Table 6).

In the first day, the content of carbonyls was 1.90 and 0.47 (nmol/mg) in the RF and PF, respectively. Carbonyl content increased up to 4.38 and 3.94 (nmol/mg) after 10 months storage at −20 °C. The amount of protein carbonyl enhanced in the period of frozen storage in fillets from both rearing systems and showed significant differences between t = 0 and t = 4. A difference was also observed in between the rearing systems at the begin of the storage time, while after 8 and 10 months we did not observe any differences between the RF and PF. When stored at +4 °C, the RF showed higher carbonyls content than PF at all times, and after 12 days of storage, the protein carbonyl content increased to 3.01 and 1.28 (nmol/mg) in the RF and PF, respectively. During post-mortem analysis, we observed a considerable increase in the total content of protein carbonyl after 8 days in fillets from both systems.

The content of MDA in the fillets from both rearing systems increased significantly (*p* < 0.05) during the 10 months of frozen storage (−20 °C) and 12 days of refrigerated storage at +4 °C (Table 5 and Table 6). The content of MDA was 0.28 MDA (µg/g) on the slaughtering day and enhanced to 0.73 and 0.69 MDA (µg/g) after 10 months and to 0.36 and 0.32 µg/g MDA after 12 days in RF and the PF, respectively. While comparing MDA values at each time point, no statistical differences were observed in the RF and PF during both storage conditions. Additionally, the level of MDA increased significantly following 4 months frozen storage, while during refrigerated storage, a significant increase was observed after 12 days.

### 3.8. Proteomic Changes during Storage at −20 °C

The protein patterns of the fillets from RAS and pond systems during the 10 months of frozen storage at −20 °C were identified with SDS-PAGE analysis. The pattern indicated several bands from 15 to 220 kDa in the fillets from both systems. The bands that appeared were suggested to be myosin heavy chain (MHC) at 200 kDa, nebulin (107 kDa), actin (43 kDa), troponin (30 kDa), and myosin light chain components (25–15 kDa) in line with [35]. Beta-enolase (49 kDa), actin (43 kDa), GAPDH (37 kDa), LDH A-chain (37 kDa), and trisephosphate isomerase (28 kDa) were confirmed by MS analysis (Appendix A).

The SDS-PAGE protein profiles indicated less intense protein bands for some of the proteins during frozen storage with elapsing time. At t = 0–4 month, the PF revealed more intense bands than RF. For both systems the most proteins band disappeared or fainted during 10 months of storage at −20 °C. MHC, α-actin, actin, and MLC showed the highest degree of fainting during storage for both systems (Figure 5a). Bands of serum albumin and troponin bands fainted less, and only after 8 and 10 months of storage.

Western blot (immunoblot) of protein carbonyl groups in the RF and PF indicated that highly intense bands at 40 and 43 kDa corresponding to actin and a non-identified intensive protein, respectively, and less intense bands at 37 kDa, which increased during frozen storage (Figure 5b). Less oxidized carbonyls were observed in PF compared to the RF at t = 0 and 4 months. Additionally, after 10 months RF showed higher content of oxidized carbonyls compared to the PF at the same time point. Results showed the progress of protein oxidation in RF and PF in the period of frozen storage.

Concomitantly, some high molecular polymers (protein aggregates, more than 200 kDa weigh) that stacked on the 1D gel, were observed by Native PAGE. Analysis of the muscle proteins of fillets from both systems by native gel electrophoresis showed highly intensive bands around 1230 kDa at t = 0 month. The band of this high molecular weight protein was very weak after 8–10 months of storage in the PF, whereas the highest loss in RF was observed after 10 months; thus, it was still quite intense after 8 months of storage in the RF. PF also showed a less intense band at 720 kDa after 8–10 months, whereas no loss was observed in the RF after storage at −20 °C. The intensity of the weak band observed around 480 kDa decreased in the PF after 8–10 months of storage, whereas no changes were observed for the RF. The band at 25 kDa was faded after 8–10 months of storage for the PF, whereas only small changes were detected for the RF over time of storage with a small decrease in band intensity after 10 months of storage (Figure 6). Comparing the results between the RF and PF revealed more stability of proteins in RF during frozen storage.

### 3.9. Principal Component Analysis Plot (PCA)

The PCA correlation loadings plot was performed to show the correlation among the chosen measured parameters during fridge storage (oxidation, hardness, pH, rigor mortis, color, biogenic amines, and liquid loss) (Figure 7a,b), as well as the chosen measured parameters during frozen storage (oxidation, hardness, proteomics study, and liquid loss) (Figure 8a,b).

PC 1 explained 77% of the variation, whereas PC 2 explained 12% for refrigerated storage. A clear correlation of the carbonyls with rigor mortis and hardness, and a negative correlation with pH and liquid loss, were demonstrated. TBARS was less correlated to the mentioned factors. More biogenic amine seemed to be formed in PF than in RF over time of storage.

For frozen storage, PC 1 explained 58% of the variation, whereas PC 2 explained 31% (Figure 8a). The carbonyl content during frozen storage correlated with liquid loss and TBARS, while hardness was negatively correlated to those parameters. Hardness was connected to MHC, MLC, nebulin, and actin. Some proteins such as MHC, MLC, nebulin, and actin, which lost their intensity during frozen storage, are connected to hardness.

Additionally, considering RF and PF according to the time progress on the PCA plot, T = 0 and 4 are located on the left side, while T = 8 and 10 are located on the right side of the plot during frozen storage. This means that the progress of time affects the examined parameters. According to this information, liquid loss, TBARS, and carbonyl content increased, while hardness and protein intensity decreased over the time in fillets from both systems.

## 4. Discussion

The fillet quality of perch from RAS and pond systems has been reported before based on sensory and textural factors [8]. However, the quality of fillets from RAS- and pond-reared fish has not been investigated in depth and not correlated with protein chemical changes caused by oxidation. In the present study, the stability and proteomic changes and the more traditional lipid oxidation were investigated in fillets from the Eurasian perch reared in two different systems.

We did not observe any difference in the fat content of the fillets from in-between the two rearing systems, whereas the fatty acid profiles indicated significant differences. In line with our results, [8,36] reported a significantly higher amount of mono-unsaturated fatty acids (MUFA) in the fillet from cultivated pike perch and intensive farmed perch compared to fillets from a pond system and higher levels of PUFA in wild perch compared to the cultivated perch. Based on the current study, RF were a better source of oleic, linoleic acids, and docosahexaenoic compared to PF. The difference might be related to the different feed ingredients. The high content of vegetable oil supplement in the RAS feed was observed in a higher content of MUFA in RF. Furthermore, the Ʃn-3: Ʃn-6 ratio did not show a difference between groups. The lipid quality indexes AI and TI were similar (below 1) in RF and PF, which indicates a very good nutritional value for human health [37].

We used PCA for the analysis of frozen and refrigerated storage separately to investigate and visualize the correlation between the evaluated parameters in depth.

Autolytic changes in perch fillets during storage at +4 °C were monitored by pH change and rigor index. Formation of lactic acid might be the main reason for the reduction of pH in both fillets until 24 h storage. On the other hand, a significantly higher pH after 72 h storage in PF than RF can be related to microbial activities and accumulation of ammonia. Furthermore, the upper situation of pond in the PCA plot compared to RAS confirmed a higher pH in the PF rather than RF over the storage time.

The onset of rigor mortis in the RF was faster than PF; also, RF in this study reached full rigor mortis 12 h earlier than PF. Moreover, we observed a better correlation between RF and rigor mortis compared to PF owing to the closer situation in the PCA plot. In fillets from both rearing systems, rigor mortis started fast due to the fish size. However, the differences between the onset of rigor mortis in the RF and PF might be related to the different age and accumulated water temperature. [38] observed that the onset of rigor mortis in carp, which were collected from cold water, had a delay compared to those accumulated from warmer water. pH and rigor index results did not show any correlation to each other, as was also confirmed with PCA analysis in this study.

Biogenic amines (BAs) in foods are generated by bacterial decarboxylation from the corresponding amino acids. Proteases are responsible in order to enhance the level of free amino acids in fish and other sea foods, especially with increasing the storage time. The content of polyamines such as SPD and SPM decrease together with the deteriorating quality of flesh [39]. The availability of decarboxylases, free amino acids, and microbial growth can be considered as important factors for the number of BAs [40].

Except for histamine and tyramine, currently there are no suggestions about critical levels of (PUT), (CAD), (SPD), and (SPM) for human consumption [21]. In accordance with the US Food and Drug Administration (FDA), the acceptable amount of histamine in fish flesh is 50 mg kg-1 ((USFDA), 1996) and the samples that were stored at +4 °C from both systems did not reach this critical amount. In total, 100–800 mg kg-1 of tyramine consumption is admissible for adults, and more than 1080 mg kg-1 is considered to be toxic [40]. The positive correlation between biogenic amine and pH found in the PCA plot indicated that pH increased with microbial activities [41]. Furthermore, the increasing amount of BAI in both fillets during storage time indicated deterioration of the fillets. In this case, the BAI for RF was 9.01 and for PF it was 7.5 after 28 days of refrigerated storage. However, both RF and PF were less than 10, which indicates acceptability of the fillet from both rearing systems at the longest storage time. The higher biogenic index in RF compared to PF might be related to the lower pH of RF. [42] reported that the activity of amino acid decarboxylase is higher in the acidic environment. Moreover, protein degradation was faster in RF compared to PF, which is shown by the amount of carbonyl content as well as Western blot.

Additionally, RF showed a higher loss of liquid and hardness compared to PF at each time point. In line with our results, the authors of [43] reported a higher liquid loss in the meagre (*Argyrosomus regius*) fillets that were reared in a tank system compared to the meagre fish fillets from a cage system. A higher liquid loss and hardness in the RF compared to the PF might be related to higher water temperature and lower swimming activity in the RAS system [43].

PF had a darker and reddish color, while RF showed a brighter color under both storage conditions. The difference in color was visually distinguishable. In line with this observation, fish reared in the pond system showed a higher a* value than RAS, which indicate more redness. This might be related to the consumption of zooplankton and zoobenthos in the pond system that may increase the redness of fish flesh [6].

Furthermore, we observed higher TBARS and carbonyl content in RF compared to PF due to the higher consumption of natural antioxidants by fish that were reared in a pond system, which gives them higher stability against oxidation progress.

During frozen storage, carbonyl content correlated with liquid loss and TBARS, while hardness was negatively correlated to those parameters (Figure 8). A positive correlation between TBARS and carbonyls during frozen storage has been reported by [44] in fish muscle. Probably, the formed products of oxidized proteins and lipids can further boost oxidation. Moreover, the products of lipid and protein oxidation considerably increased after 4 months of storage at −20 °C in fillets from both rearing systems, which confirms that protein and lipid oxidation probably started together in both types of fillets. On the contrary, less correlation was found between TBARS and carbonyl content during refrigerated storage (Figure 7). However, a noticeable lipid and protein oxidation development was observed after 12 and 8 days, respectively. Probably, protein oxidation initiated earlier than lipid oxidation during refrigerated storage, which was recently indicated by [45].

We used the proteomic study to identify changes in the protein pattern in RF and PF over time of frozen storage. We found changes by elapsing time in both systems contrary to [46], who observed similar total protein profiles when comparing fillets from wild sea and farmed breams on the day of slaughtering. Declined intensity in actin and myosin bands was very pronounced in the fillets of both rearing systems in our results with respect to elapsed time. At the same time, more protein carbonyls were observed in the RF compared to PF, confirmed both by Western blotting and spectrophotometric analysis. Storage increased the protein oxidation in fillets from both systems, while more protein oxidation was detected by Western blot in RF with respect to protein carbonyls, especially, at the early stage of storage. [47] reported that protein structure is important to extend protein oxidation, with lose protein structures giving higher oxidative changes. Even though the proteins in fish are highly structured with disulfide bands, we propose that the protein structure might differ between the RF and PF. More well-defined structured proteins are more shielded against oxidation in the period of storage [48]. In the case of our results, PF probably holds a tighter structure than RF indicated by the lower hardness of PF rather than RF after 10 months of storage. This might be another reason for what we mentioned above.

The positive correlation between hardness and specified proteins (MHC, MLC, nebulin, and actin) and the negative correlation with the amount of carbonyl measured by UV spectrophotometric method means that carbonyl content increased with elapsing time when the intensity of specified proteins and hardness decreased in both rearing systems. We observed a major reduction in hardness after 4 months of frozen storage in PF and RF (43% and 40%, respectively), while during the remaining storage time we also observed this reduction percentage. The lower level of protein carbonylation in t = 0 and t = 4 than in t = 8 and t = 10, alongside the higher hardness reduction in the first 4 months of storage, indicated that the formation of ice crystals followed by cell disruption can be considered as a key role in the hardness deterioration rather than protein denaturation during first months of frozen storage [49], probably, formation of large ice crystal due to the slow freezing speed can be considered as a key role for both fillets deterioration while, in the rest of storage time, protein denaturation and cross-linking were dominant.

On the other hand, a positive correlation between hardness and carbonyls during refrigerated storage confirmed the effect of protein disruption on the fillets’ deterioration from both rearing systems. However, the hardness reduction is probably caused by proteolytic activity [50], due to a less rigid structure that may favor the oxidation [51]. Furthermore, liquid loss showed an increasing trend over time under both storage conditions [9,52]. A high correlation between liquid loss and carbonyl in frozen storage and low negative correlation in refrigerated storage was confirmed by PCA. We observed the same increasing trend in liquid loss and carbonyls during each time point of frozen storage. It suggested that a higher degree of cross-linked proteins due to protein denaturation might increase liquid loss, while, during refrigerated storage, proteolytic enzyme activity may be the main reason.

## 5. Conclusions

This study indicated an impact of rearing conditions on fish fillet quality and the progress of lipid and protein oxidation during short- and long-term storage. During the refrigerated storage, protein oxidation seemed to initiate the oxidation rather than lipid oxidation. RF showed higher liquid loss, hardness, and protein oxidation and less native protein than PF suggesting that protein oxidation and cross-linkage of proteins in the RF were highly important for fillet quality. Optimizing the amount of antioxidants in the fish diet reared in RAS can be considered for further studies. Furthermore, during the first months of frozen storage, the formation of ice crystals may be considered as the main factor for the textural deterioration. It is notable that the amount of pH and rigor index showed faster autolytic changes in RF compared to the PF. Additionally, the TI, AI, and n-3/n-6 ratio indicated almost the same nutritional quality in both fillets. Furthermore, we proposed that the consumption of a different diet influenced color as well as other quality parameters in the current study. Therefore, concerning the final quality of the fillets from both rearing systems, we observed a better oxidation stability and lower BAI in the fillets from the pond systems.

## Figures and Tables

**Figure 1 foods-10-01405-f001:**
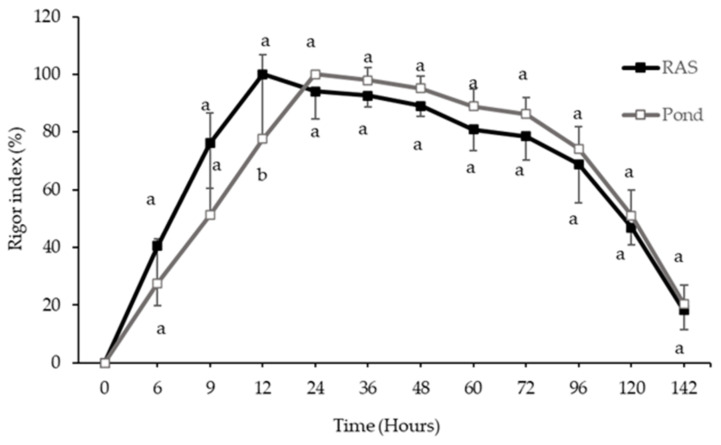
Changes of rigor index of perch from recirculating aquaculture systems (RAS) and pond during 140 h refrigerated storage at +4 °C. Different small superscript letters on the line indicate significant difference (*p* < 0.05) between rearing systems at the same time point (mean ± S.D., *n* = 6).

**Figure 2 foods-10-01405-f002:**
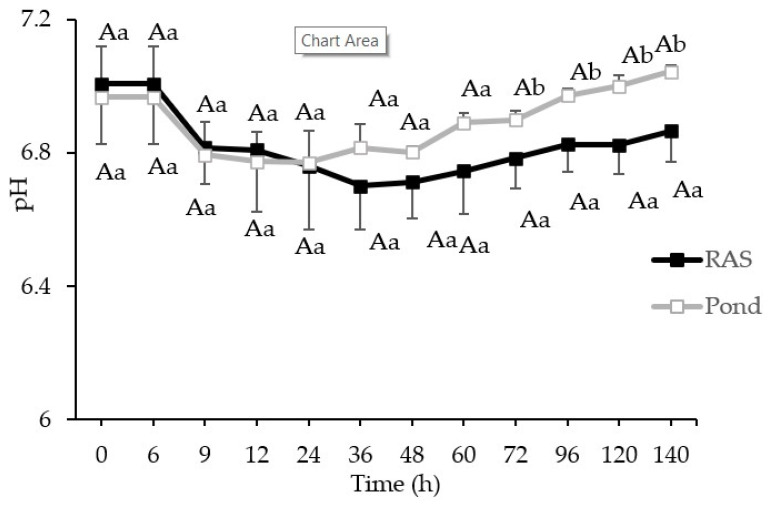
pH of perch from RAS and pond during 140 h refrigerated storage at +4 °. Different capital letters on the lines donate significant differences (*p* < 0.05) in each rearing system. Small letters on the lines donate significant differences (*p* < 0.05) between rearing systems at the similar time point (mean ± S.D., *n* = 6).

**Figure 3 foods-10-01405-f003:**
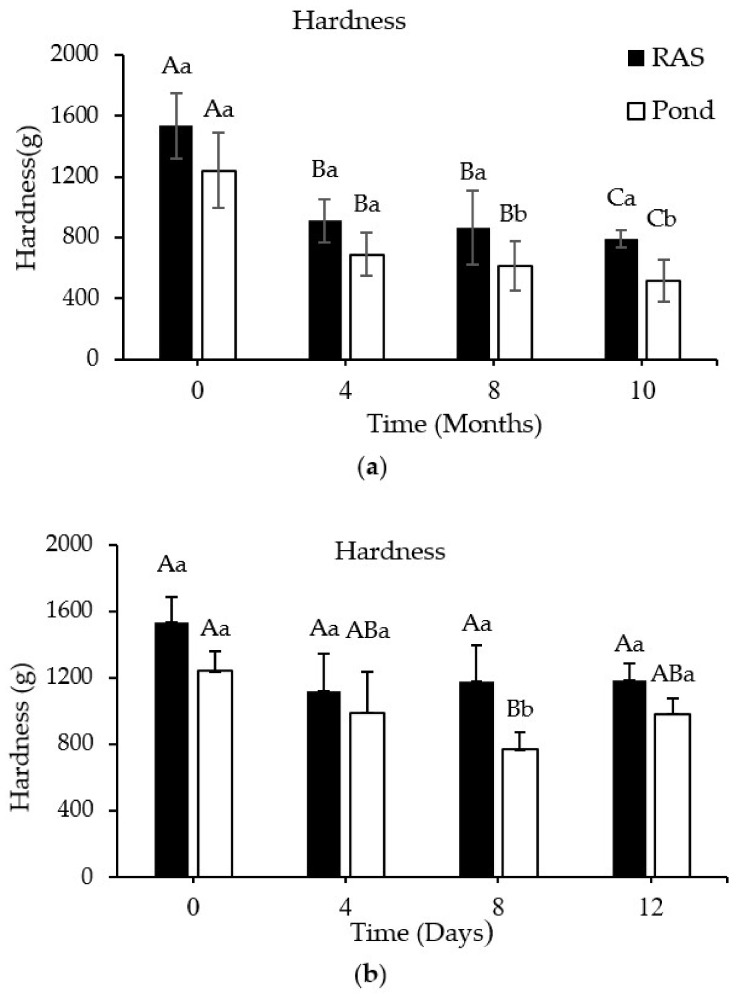
(**a**) Changes in hardness of perch fillets from RAS and pond systems during storage at –20 °C and (**b**) storage at +4 °C (mean ± S.D., *n* = 6), respectively. Different capital letters on the bar donate significant differences (*p* < 0.05) within each rearing system. Small superscript letters on the bar donate significant differences (*p* < 0.05) between rearing systems at the same time point.

**Figure 4 foods-10-01405-f004:**
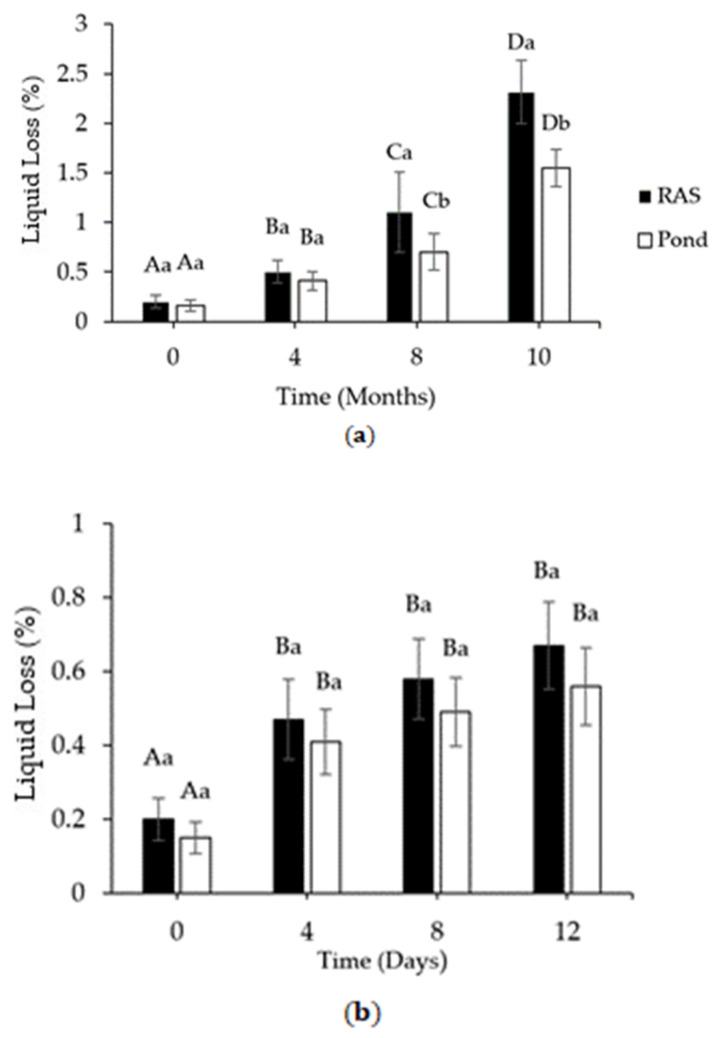
(**a**) Changes in liquid loss of perch fillets from RAS and pond during storage at −20 °C and (**b**) storage at +4 °C (mean ± S.D., *n* = 6), respectively. Different capital letters on the bar donate significant differences (*p* < 0.05) within each rearing system. Small letters on the bar donate significant differences (*p* < 0.05) between rearing systems at the same time point.

**Figure 5 foods-10-01405-f005:**
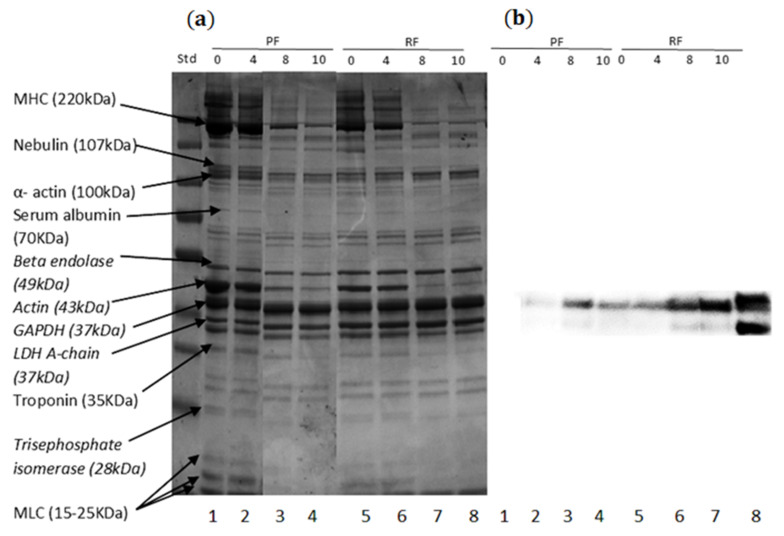
(**a**) SDS-polyacrylamide gel electrophoresis, (**b**) immunoblotting against protein carbonyl groups in perch fillet from pond (line 1–4) and RAS (line 5–8) systems during frozen storage at −20 °C. Line (1) t = 0, pond fillet; (2) 4 months, pond fillet; (3) 8 months, pond fillet; and (4) 10 months; (5) at t = 0, RAS fillet; (6) 4 months, RAS fillet; (7) 8 months, RAS fillet; and (8) 10 months, RAS fillet.

**Figure 6 foods-10-01405-f006:**
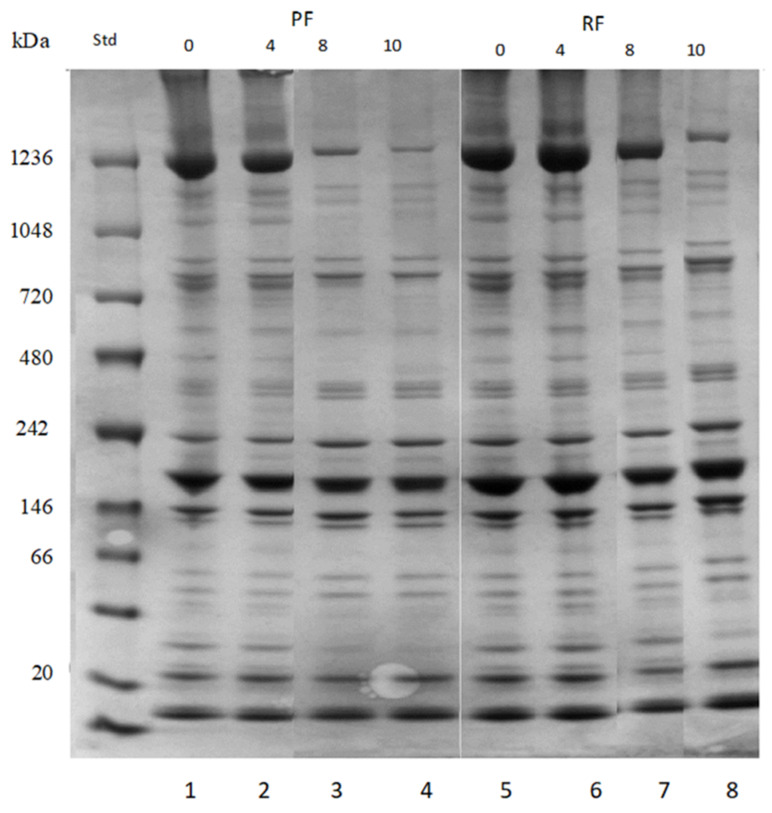
Native PAGE, Bis-Tris gel of perch from the pond (line 1–4) and RAS (line 5–8) systems during frozen storage at −20 °C. Line (**1**) t = 0, PF; (**2**) 4 months, PF; (**3**) 8 months, PF; and (**4**) 10 months PF; line (**5**) at t = 0, RF; (**6**) 4 months, RF; (**7**) 8 months, RF; and (**8**) 10 months, RF.

**Figure 7 foods-10-01405-f007:**
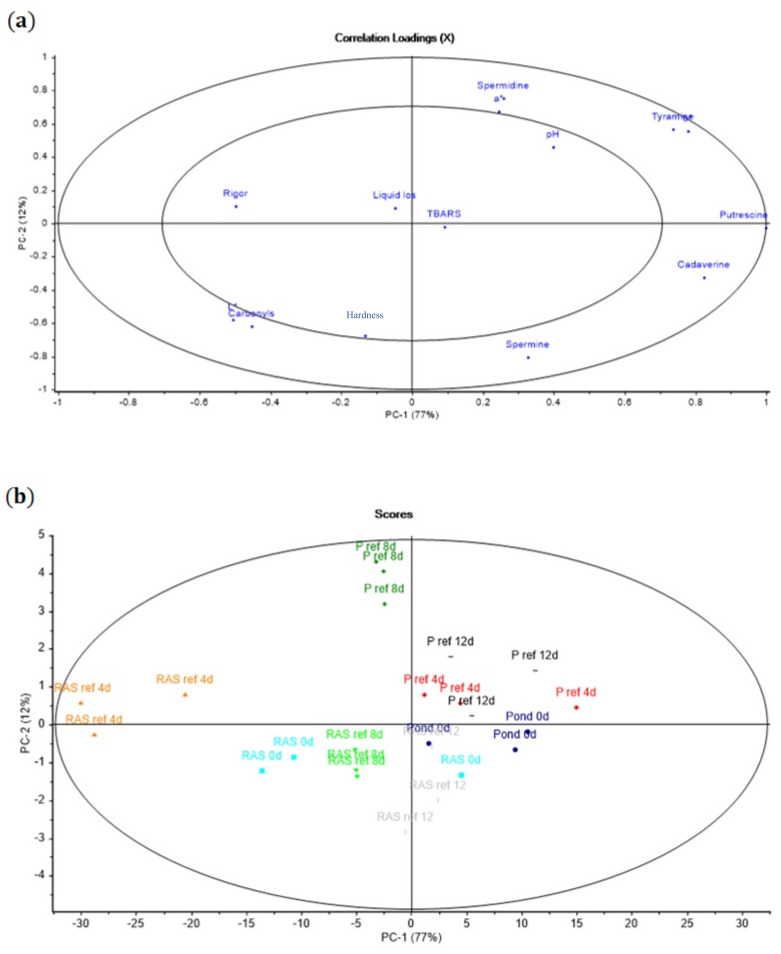
(**a**) PCA plot showing the correlation loadings on PC 1 and PC 2 of the oxidation parameters TBARS and the carbonyls pH, liquid loss, rigor mortis, color, and hardness parameters of RF and PF during refrigerated storage. (**b**) PCA plot showing the correlation loadings on PC 1 and PC 2 between the storage time for each rearing system.

**Figure 8 foods-10-01405-f008:**
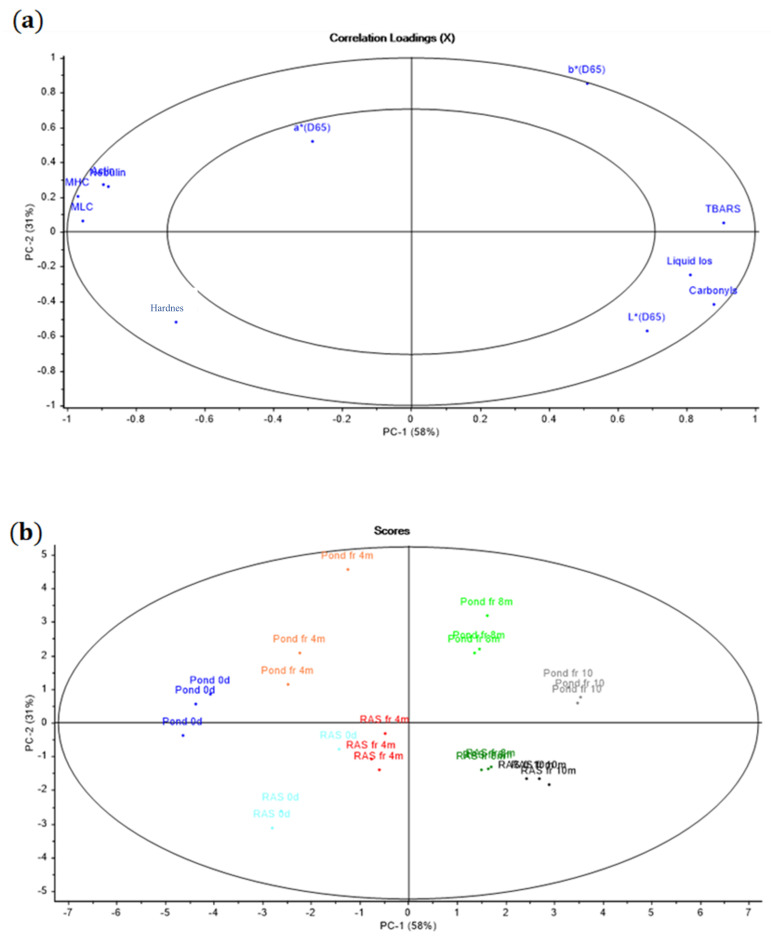
(**a**) PCA plot showing the correlation loadings on PC 1 and PC 2 of the oxidation parameters TBARS and carbonyls, liquid loss, color, specified proteins and hardness parameters of RF and PF during frozen storage. (**b**) PCA plot showing the correlation loadings on PC 1 and PC 2 between the storage time for each rearing system.

**Table 1 foods-10-01405-t001:** Fat content (%) and fatty acids composition (%) of Eurasian perch fillets from RAS and pond rearing systems (mean ± S.D., *n* = 6).

	RAS	Pond
**Fat content (%)**	1.19 ± 0.23 ^a^	1.09 ± 0.31 ^a^
**Fatty acid (%)**	**RAS**	**Pond**
C14:0	2.14 ± 0.22 ^a^	1.62 ± 0.11 ^b^
C16:0	20.8 ± 0.47 ^a^	20.38 ± 0.79 ^a^
C18:0	2.8 ± 0.33 ^a^	3.63 ± 0.41 ^b^
**SFA**	**25.85 ^a^**	**25.72 ^a^**
C14:1	0.24 ± 0.04 ^a^	0.14 ± 0.03 ^b^
C16:1	6.53 ± 0.39 ^a^	8.15 ± 1.43 ^b^
C18:1n-9	19.26 ± 1.11 ^a^	13.38 ± 1.58 ^b^
C18:1n-7	2.68 ± 0.21 ^a^	4.47 ± 0.38 ^b^
C20:1n-9	1.78 ± 0.21 ^a^	0.36 ± 0.06 ^b^
**MUFA**	**30.85 ^a^**	**26.56 ^b^**
C18:2n-6	7.19 ± 0.52 ^a^	3.96 ± 0.55 ^b^
C18:3n-3	1.32 ± 0.11 ^a^	4.37 ± 0.20 ^b^
C20:3n-3	0.16 ± 0.03 ^a^	0.48 ± 0.08 ^b^
C20:4n-6	1.29 ± 0.14 ^a^	7.12 ± 0.95 ^b^
C20:5n-3	4.89 ± 0.38 ^a^	6.52 ± 0.70 ^b^
C22:5n-3	1.08 ± 0.10 ^a^	2.66 ± 0.13 ^b^
C22:6n-3	26.95 ± 1.77 ^a^	22.18 ± 1.04 ^b^
**PUFA**	**43.2 ^a^**	**47.62 ^b^**
n-3	34.4 ± 11.36 ^a^	29.99 ± 8.64 ^a^
n-6	8.8 ± 3.71 ^a^	11.41 ± 3.39 ^a^
n-3/n-6	3.90 ^a^	2.62 ^a^
TI	0.39 ± 0.02 ^a^	0.39 ± 0.02 ^a^
AI	0.22 ± 0.01 ^a^	0.22 ± 0.01 ^a^

Different small letters in a row indicate significant difference (*p* < 0.05) between rearing systems. RAS, recirculating aquaculture systems. SFA, saturated fatty acids. MUFA, monounsaturated fatty acids. PUFA, polyunsaturated fatty acids.

**Table 2 foods-10-01405-t002:** Content of biogenic amines (mg/kg) in fillets of Eurasian perch from RAS and pond rearing systems (mean ± SD; *n* = 5) stored at +4 °C.

Content of Biogenic Amines	Time (Days)	7	14	21	28
Putrescine	-	-	-	-	-
RAS	(4 °C)	ND ^A,a^	4.6 ± 2.30 ^A,b^	26.3 ± 6.15 ^A,c^	51.6 ± 15.6 ^A,d^
Pond	(4 °C)	1.7 ± 0.48 ^B,a^	10.6 ± 2.89 ^B,b^	33.2 ± 2.23 ^A,c^	41.7 ± 14.3 ^A,d^
Cadaverine					
RAS	(4 °C)	ND ^A,a^	2.3 ± 0.15 ^A,b^	17.6 ± 9.56 ^A,c^	27.9 ± 14.8 ^A,d^
Pond	(4 °C)	ND ^A,a^	2.4 ± 0.66 ^A,b^	9.1 ± 2.58 ^A,c^	13.8 ± 10.6 ^A,d^
Tyramine	-	-	-	-	-
RAS	(4 °C)	1.4 ± 0.21 ^A,a^	1.7 ± 0.72 ^A,a^	8.8 ± 3.13 ^A,b^	16.4 ± 6.10 ^A,c^
Pond	(4 °C)	2.3 ± 0.57 ^B,a^	3.9 ± 0.80 ^B,b^	12.5 ± 4.68 ^A,c^	16.7 ± 5.15 ^A,d^
Histamine	-	-	-	-	-
RAS	(4 °C)	ND ^A,a^	ND ^A,a^	ND ^A,a^	0.72 ± 0.45 ^A,b^
Pond	(4 °C)	ND ^A,a^	ND ^A,a^	ND ^A,a^	0.68 ± 0.17 ^A,b^
Spermidine	-	-	-	-	-
RAS	(4 °C)	2.2 ± 0.38 ^A,a^	2.4 ± 0.52 ^A,a^	2.2 ± 0.17 ^A,a^	1.8 ± 0.23 ^A,a^
Pond	(4 °C)	3.0 ± 0.66 ^A,a^	3.2 ± 0.30 ^A,a^	2.4 ± 0.89 ^A,a^	2.0 ± 0.36 ^A,a^
Spermine	-	-	-	-	-
RAS	(4 °C)	6.6 ± 1.04 ^A,a^	7.1 ± 0.78 ^A,a^	7.2 ± 0.68 ^A,a^	6.1 ± 1.23 ^A,a^
Pond	(4 °C)	4.8 ± 0.36 ^B,a^	7.0 ± 0.41 ^A,b^	4.2 ± 0.49 ^B,a^	4.4 ± 1.21 ^A,a^
**Biogenic Amines Index (BAI)**	-	-	-	-	-
RAS	(4 °C)	0	0.6	4.2	9.01
Pond	(4 °C)	0.1	1.1	5.5	7.5

Different capital letters in the columns donate significant differences (*p* < 0.05) in each rearing system. Small letters in the columns donate significant difference (*p* < 0.05) between rearing systems at the same time point. ND: not detected.

**Table 3 foods-10-01405-t003:** Color changes in fish fillets from RAS and pond rearing systems during 10 months of frozen storage at −20 °C (mean ± S.D., *n* = 6).

Colour	Time (Months)	RAS	Pond
Lightness (L*)	0	47.22 ± 2.2 ^A,a^	44.26 ± 1.57 ^A,b^
	4	48.08 ± 1.89 ^A,a^	43.12 ± 1.40 ^A,b^
	8	51.55 ± 2.15 ^B,a^	45.86 ± 1.39 ^A,b^
	10	53.67 ± 1.95 ^C,a^	49.10 ± 1.45 ^B,b^
Redness (a*)	0	−2.62 ± 0.28 ^A,a^	2.04 ± 0.59 ^A,b^
	4	−2.79 ± 0.47 ^A,a^	−1.85 ± 0.79 ^A,b^
	8	−2.94 ± 0.39 ^B,a^	−2.34 ± 0.84 ^B,b^
	10	−3.27 ± 0.5 ^C,a^	−2.69 ± 0.57 ^C,b^
Yellowness (b*)	0	3.29 ± 0.71 ^A,a^	5.74 ± 0.78 ^A,b^
	4	4.42 ± 1.36 ^B,a^	6.43 ± 1.73 ^B,b^
	8	4.53 ± 0.76 ^B,a^	7.2 ± 1.1 ^C,b^
	10	5.03 ± 1.17 ^C,a^	7.44 ± 1.07 ^C,b^

Different capital letters in a column donate significant differences (*p* < 0.05) within each rearing system. Small letters in a row donate significant difference (*p* < 0.05) between rearing systems at the same time point.

**Table 4 foods-10-01405-t004:** Color changes in fish fillets from RAS and pond rearing systems during 12 days of refrigerated storage at + 4 °C (mean ± S.D., *n* = 6).

	Time (Months)	RAS	Pond
Lightness (L*)	0	47.22 ± 2.2 ^A,a^	44.26 ± 1.57 ^A,b^
	4	46.2 ± 1.69 ^A,a^	43.19 ± 2.89 ^A,b,b^
	8	46.64 ± 1.49 ^A,a^	40.99 ± 2.34 ^B,b^
	10	46.66 ± 4.54 ^A,a^	42.45 ± 2.25 ^A,b,b^
Redness (a*)	0	−2.62 ± 0.28 ^A,a^	−2.04 ± 0.59 ^A,b^
	4	−2.63 ± 0.47 ^A,a^	−2.63 ± 0.79 ^A,b^
	8	−2.3 ± 0.39 ^A,a^	−1.96 ± 0.84 ^A,a^
	10	−2.69 ± 0.5 ^A,a^	−1.89 ± 0.57 ^A,b^
Yellowness (b*)	0	3.29 ± 0.71 ^A,a^	5.74 ± 1.59 ^A,b^
	4	2.46 ± 1.28 ^B,a^	3.78 ± 0.86 ^B,b^
	8	2.54 ± 0.99 ^B,a^	3.78 ± 0.86 ^B,b^
	10	3.45 ± 1.17 ^A,a^	5.04 ± 1.07 ^A,b,b^

Different capital letters in a column donate significant differences (*p* < 0.05) within each rearing system. Small letters in a row donate significant difference (*p* < 0.05) between rearing systems at the same time point.

**Table 5 foods-10-01405-t005:** MDA (µg/g) and carbonyl content (nmol/mg) parameters in perch fillets from RAS and pond rearing systems during frozen storage at −20 °C (mean ± S.D., *n* = 6).

Time	MDA (µg/g)		Protein Carbonyls (nmol/mg)	
Months	RAS	POND	RAS	POND
0	0.28 ± 0.01 ^A,a^	0.28 ± 0.0 ^A,a^	1.9 ± 0.01 ^A,a^	0.47 ± 0.00 ^A,b^
4	0.51 ± 0.01 ^B,a^	0.48 ± 0.02 ^B,a^	3.1 ± 0.00 ^B,a^	1.78 ± 0.01 ^B,b^
8	0.56 ± 0.05 ^B,a^	0.53 ± 0.02 ^B,a^	3.82 ± 0.02 ^B,a^	3.1 ± 0.02 ^C,a^
10	0.73 ± 0.04 ^C,a^	0.69 ± 0.02 ^C,a^	4.38 ± 0.03 ^C,a^	3.94 ± 0.03 ^C,a^

Different capital letters in a column donate significant differences (*p* < 0.05) within each rearing system. Small letters in a row donate significant difference (*p* < 0.05) between rearing systems at the same time point.

**Table 6 foods-10-01405-t006:** MDA (µg/g) and carbonyl content (nmol/mg) parameters in perch fillets from RAS and pond rearing systems during refrigerated storage at +4 °C (mean ± S.D., *n* = 6).

Time	MDA (µg/g)		Protein Carbonyls (nmol/mg)	
Months	RAS	POND	RAS	POND
0	0.28 ± 0.01 ^A,a^	0.28 ± 0.0 ^A,a^	1.9 ± 0.01 ^A,a^	0.47 ± 0.00 ^A,b^
4	0.29 ± 0.01 ^A,a^	0.28 ± 0.02 ^A,a^	2.24 ± 0.02 ^A,a^	0.53 ± 0.03 ^A,b^
8	0.34 ± 0.05 ^A,B,a^	0.31 ± 0.02A ^B,a^	2.71 ± 0.03 ^B,a^	0.9 ± 0.01 ^B,b^
10	0.36 ± 0.04 ^B,a^	0.32 ± 0.02 ^B,a^	3.01 ± 0.03 ^C,a^	1.28 ± 0.05 ^B,b^

Different capital letters in a column donate significant differences (*p* < 0.05) within each rearing system. Small letters in a row donate significant difference (*p* < 0.05) between rearing systems at the same time point.

## Data Availability

Not applicable.

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
