# Peer review of "Comparison of Quality Changes in Eurasian Perch (Perca fluviatilis L.) Fillets Originated from Two Different Rearing Systems during Frozen and Refrigerated Storage"

_foods, 2021, doi:10.3390/foods10061405_

Round 1
Reviewer 1 Report
The authors have conducted considerable work on this project. However, considerable re-evaluation of this manuscript is needed.
A description of the freezing/cooling procedures, packaging, and storage conditions should be included in this manuscript.
Abstract:
In the Abstract the authors state: “A good correlation between carbonyls, TBARS and liquid loss with negative correlation with firmness and specified proteins during frozen storage confirmed that freezing is the main reason for fillet deterioration rather than protein oxidation….” However, much of the textural issues in this manuscript maybe the result of the incorrect freezing/cooling procedures, poor packaging and storage conditions. If the perch fillets were placed into a storage freezer with still air at -18C or in a refrigerated room at +4C with no moving air, and not ensuring that the fillets are separated, it will require considerable time, especially for the frozen fillets, to be reduced in temperature. Slow freezing of the fillets will result in large ice crystal formation in the flesh, which can result in oxidation, loss of color, textural cross-linkage, etc. If the fish are slow frozen in a storage freezer, it may take several days if not weeks to reduce the temperature from +20C to -18C. This slow freezing process will result in large ice crystals, since, at -7C and -10C approximately 24%, and 16%, respectively, of the moisture is still unfrozen, and the salts and minerals present in the fish flesh are dissolved in the unfrozen moisture, reducing the freezing point. This results in large ice crystals in the flesh, causing textural cross linkage, lipid oxidation, color loss, etc.
There is also a Temperature Accelerating Effect (Q10). This is the effect of temperature on the chemical or biochemical reaction rate. The Chemical reaction rate increases 2-3-fold with flesh temperature is increased by 10C. Similarly, the chemical reaction rate decreases 2-3-fold when flesh temperature is decreased by 10C. This effects both the refrigeration and frozen process.
Materials and Methods:
Many of the differences between the PF and RF fish in this manuscript are also due to the diet and not inherently due to whether or not these fish were pond raised or aqua-cultured. The authors should have analyzed the zooplankton and algae in the ponds for proximate composition and fatty acid profiles. They also need to provide a more detailed description of the nutritional and fatty acid profiles of the feed for the RAS fish.
Rigor Index
The sizes of the fillets and temperature of the individual fillets should have been included in the reported data. The average weight for the PF was approximately 28g higher compared with the RA fish. It is possible that the temperature of the PF fish was also slightly higher due to its larger size, which may have affected the time to full rigor in PF versus RF.
Statistical Analyses
The entire experiment should have been replicated. Some of the statistical analyses also need review. For example, for Figure 4A, it appears that the standard deviation bars at 8 days and 12 days overlap suggesting that there were no significant differences between rearing systems. However, the authors 2-way ANOVA showed significant differences. The authors should have also used Tukey’ Studentized Range Test to statistically analyze significant differences between the means. If they had done this on this analyses and other analyses, it would potentially have shown no significant differences. The Tukey's honestly significant difference test (Tukey's HSD) is used to test differences among sample means for significance. The Tukey's HSD tests all pairwise differences while controlling the probability of making one or more Type I errors.
Results:
Page 17, Lines 496 and 497, the authors state “More biogenic amine seemed to be formed in PF than in RF over time of storage.” The authors should explain why this occurred and its importance, if any.
Discussion:
On line 529, the authors state “The difference might be related to the different feed ingredients.” This is correct, and is the reason why such data should have been collected. On Lines 567-569, the authors write “Higher liquid loss and firmness in the RF compared to the PF might be related to higher water temperatures and lower swimming activity in the RAS system. However, the authors should have used Tukey’ Studentized Range Test to statistically analyze significant differences between the means. If they had done this on this analyses and other analyses, it would potentially have shown no significant differences. The Tukey's honestly significant difference test (Tukey's HSD) is used to test differences among sample means for significance. The Tukey's HSD tests all pairwise differences while controlling the probability of making one or more Type I errors.
Lines 614-616, the authors write “formation of ice crystals followed by cell disruption can be considered as a key role in the firmness deterioration rather than protein denaturation during the first months of frozen storage….” This is correct, the cause of the large ice crystals and cell disruption maybe due to the fact that these fillets were slow frozen rather than quick (i.e. blast) freezing (See Above). Blast freezing will result in smaller ice crystal formation and a better-quality product especially if afterwards these fillets are properly packaged and stored at -18C.

Author Response
Point 1: Abstract:
In the Abstract the authors state: “A good correlation between carbonyls, TBARS and liquid loss with negative correlation with firmness and specified proteins during frozen storage confirmed that freezing is the main reason for fillet deterioration rather than protein oxidation….” However, much of the textural issues in this manuscript maybe the result of the incorrect freezing/cooling procedures, poor packaging and storage conditions. If the perch fillets were placed into a storage freezer with still air at -18C or in a refrigerated room at +4C with no moving air, and not ensuring that the fillets are separated, it will require considerable time, especially for the frozen fillets, to be reduced in temperature. Slow freezing of the fillets will result in large ice crystal formation in the flesh, which can result in oxidation, loss of color, textural cross-linkage, etc. If the fish are slow frozen in a storage freezer, it may take several days if not weeks to reduce the temperature from +20C to -18C. This slow freezing process will result in large ice crystals, since, at -7C and -10C approximately 24%, and 16%, respectively, of the moisture is still unfrozen, and the salts and minerals present in the fish flesh are dissolved in the unfrozen moisture, reducing the freezing point. This results in large ice crystals in the flesh, causing textural cross linkage, lipid oxidation, color loss, etc.
There is also a Temperature Accelerating Effect (Q10). This is the effect of temperature on the chemical or biochemical reaction rate. The Chemical reaction rate increases 2-3-fold with flesh temperature is increased by 10C. Similarly, the chemical reaction rate decreases 2-3-fold when flesh temperature is decreased by 10C. This effects both the refrigeration and frozen process.
Response 1: We agree with all your statements regarding ice crystal formation and freezing speed. However, we would like to mention that the slaughtering done in a cold room (+4 oC) then, fillets were washed in a child water (0 oC). Additionally, all the processes such as filleting, packaging and storage were done in the same way for all fillets. As we compared the fillet quality of RAS and pond in this study by the effect of frozen storage the development and formation of ice crystal should be the same in both fillets. Therefore, we would expect to have an equal impact of freezing on fillets from both rearing systems. Furthermore, we tried to make a simulation of domestic freezers which are mostly used and consider the influence of slow freezing on the quality of perch fillets. We have already published a paper about frozen storage in plastic pack on common carp (Cyprinus carpio L.). Please find the references in the following sentence: Hematyar, N., et al., Nutritional quality, oxidation, and sensory parameters in fillets of common carp (Cyprinus carpio L.) influenced by frozen storage (–20 °C). Journal of Food Processing and Preservation, 2018. 42(5): p. e13589.
Point 2: Materials and Methods:
Many of the differences between the PF and RF fish in this manuscript are also due to the diet and not inherently due to whether or not these fish were pond raised or aqua-cultured. The authors should have analyzed the zooplankton and algae in the ponds for proximate composition and fatty acid profiles. They also need to provide a more detailed description of the nutritional and fatty acid profiles of the feed for the RAS fish.
Response 2: Thank you for your constructive comment. In fact, perch starts to be piscivores from size of 10 cm approximately they feed mostly on small fish size. Therefore, we did not consider zooplankton and algae as the main concern of fillet quality. Concerning the different feeds in the two rearing systems that is exactly what makes the difference between the systems. Please find proximate composition and fatty acid profiles of zooplankton in tables 1and 2. Please find the nutritional and fatty acid profiles of the feed for the RAS fish in table 3. Biomar (EFICO Sigma 970 3 mm).
Table1.
|
Fatty acids content (% of DM) |
||
|
Fatty acid |
CYCLOPS |
DAPHNIA |
|
C8_0 |
0,00 |
0,00 |
|
C10_0 |
3,87 |
0,00 |
|
C11_0 |
0,00 |
0,00 |
|
C12_0 |
4,12 |
6,17 |
|
C13_0 |
1,13 |
2,11 |
|
C14_0 |
117,23 |
190,18 |
|
C14_1 |
16,53 |
43,72 |
|
C15_0 |
8,78 |
22,85 |
|
C15_1 |
3,05 |
7,11 |
|
C16_0 |
253,19 |
468,91 |
|
C16_1 |
258,87 |
302,15 |
|
C16_2 |
0,00 |
0,00 |
|
C17_0 |
10,91 |
18,47 |
|
C17_1 |
3,96 |
6,05 |
|
C16_3 |
11,37 |
7,72 |
|
C16_4 |
0,00 |
0,00 |
|
C18_0 |
44,98 |
59,31 |
|
C18_1n9trans |
1,47 |
0,00 |
|
C18_1n9 |
43,65 |
126,86 |
|
C18_1n7 |
45,22 |
59,11 |
|
C18_2n6_trans |
1,56 |
0,00 |
|
C18_2n6 |
22,09 |
32,19 |
|
C18_3n6 |
2,19 |
1,63 |
|
C18_3n3 |
49,22 |
42,64 |
|
C20_0 |
1,29 |
0,00 |
|
C20_1n9 |
3,40 |
5,51 |
|
C20_2 |
0,00 |
0,00 |
|
C20_3n6 |
2,89 |
4,31 |
|
C21_0 |
1,06 |
0,00 |
|
C20_3n3 |
6,81 |
3,04 |
|
C20_4n6 |
3,32 |
2,39 |
|
C22_0 |
1,09 |
0,00 |
|
C22_1n9 |
1,65 |
3,27 |
|
C20_5n3 |
57,38 |
23,75 |
|
C22_2 |
3,74 |
6,37 |
|
C24_0 |
1,01 |
0,00 |
|
C24_1n9 |
3,59 |
4,67 |
|
C22_5n6 |
0,00 |
0,00 |
|
C22_6n3 |
39,39 |
9,50 |
|
SUFA |
448,65 |
768,00 |
|
MUFA |
381,40 |
558,46 |
|
PUFA |
196,22 |
127,17 |
|
n3 |
152,81 |
78,93 |
|
n6 |
32,04 |
40,51 |
|
n6/n3 |
0,21 |
0,51 |
Table 2.
|
Protein g/100g DM |
Lipid g/100 g DM |
Fibre g/100 g DM |
Ash g/100 g DM |
NFE g/100 g DM |
Energy g/100 g DM |
P g/100 g DM |
NPE kcal/100g DM |
NPE:P cal/mg DM |
||
|
Chironomid |
54,81 |
4,15 |
3,9 |
18,65 |
18,48 |
331 |
0,8 |
111,27 |
2,03 |
|
|
Zooplankton |
66 |
8,54 |
8,02 |
8,52 |
8,89 |
376 |
1,28 |
112,42 |
1,70 |
|
|
Table 3. |
|
||||||
|
FA |
BIO |
Essential AA |
BIO |
DM |
93,4 |
||
|
C12:0 |
0,09 |
Arginine |
4,5 |
CP |
53,4 |
||
|
C14:0 |
4,49 |
Histidine |
2,3 |
EE |
18,7 |
||
|
C16:0 |
16,98 |
Isoleucine |
3,8 |
Ash |
9,1 |
||
|
C16:1 |
5 |
Leucine |
7,3 |
NFE (g /100g)h |
18,8 |
||
|
C18:0 |
3,72 |
Lysine |
6,2 |
Gross energy (MJ /kg)i |
20,62 |
||
|
C18:1n9 |
16,6 |
Methionine |
1,9 |
||||
|
C18:1n7 |
16,61 |
Phenylalanine |
4,6 |
||||
|
C18:2n6 |
7,61 |
Tyrosine |
3,2 |
||||
|
C18:3n3 |
1,9 |
Threonine |
3,5 |
||||
|
C20:1n9 |
3,17 |
Valine |
4,2 |
||||
|
C20:3n3 |
0,18 |
Non-essential AA |
|||||
|
C20:4n6 |
0,92 |
Alanine |
4,6 |
||||
|
C20:5n3 |
0,48 |
Aspartic acid |
7,7 |
||||
|
C22:5n6 |
1,43 |
Glycine |
4,6 |
||||
|
C22:6n3 |
14,47 |
Glutamic acid |
19 |
||||
|
C23:0 |
0,99 |
Proline |
9,7 |
||||
|
SFA |
27,92 |
Serine |
4 |
||||
|
MUFA |
42,75 |
||||||
|
PUFA |
28,37 |
||||||
|
PUFA+MUFA |
71,12 |
||||||
|
n3 |
17,03 |
||||||
|
n6 |
10,35 |
||||||
|
n3/n6 |
1,65 |
||||||
|
UI |
167,41 |
||||||
|
AI |
0,49 |
||||||
|
TI |
0,27 |
Point 3: Rigor Index
The sizes of the fillets and temperature of the individual fillets should have been included in the reported data. The average weight for the PF was approximately 28g higher compared with the RA fish. It is possible that the temperature of the PF fish was also slightly higher due to its larger size, which may have affected the time to full rigor in PF versus RF.
Response 3: That is right. We fully agree that the size might have impact of rigor mortis. However, please consider that in total the average of RAS fish was 28 g higher per 30 fish compared to fish from pond not per single fish and we believe it could not make difference between rigor index. Additionally, we tried to take fish with similar size for rigor mortis.
Point 4: Statistical Analyses
The entire experiment should have been replicated. Some of the statistical analyses also need review. For example, for Figure 4A, it appears that the standard deviation bars at 8 days and 12 days overlap suggesting that there were no significant differences between rearing systems. However, the authors 2-way ANOVA showed significant differences. The authors should have also used Tukey’ Studentized Range Test to statistically analyze significant differences between the means. If they had done this on this analyses and other analyses, it would potentially have shown no significant differences. The Tukey's honestly significant difference test (Tukey's HSD) is used to test differences among sample means for significance. The Tukey's HSD tests all pairwise differences while controlling the probability of making one or more Type I errors.
Response 4: We agree and have corrected in the figures 3 and 4. RF showed higher hardness after 8 months frozen storage (fig 3). There is no significant differences between rearing systems at 8 and 12 days (fig 4). Regarding Tukey's HSD tests, we would like to inform you that we did this test for all of analysis in this manuscript. We included this information in the lines 347-348.
Point 5: Results:
Page 17, Lines 496 and 497, the authors state “More biogenic amine seemed to be formed in PF than in RF over time of storage.” The authors should explain why this occurred and its importance, if any.
Response 5: We explain the reasons which might influence on biogenic amines development in lines 688-695.
“Furthermore, increasing amount of BAI in both fillets during storage time indicated deterioration of fillets. In this case, BAI for RF was 9.01 and for PF was 7.5 after 28 days refrigerated storage. However, both RF and PF were less than 10 which indicate acceptability of fillet from both rearing systems at the longest storage time. Higher bio-genic index in RF compared to PF might be related to the lower pH of RF. [8] reported that the activity of amino acid decarboxylase is higher in the acidic environment. Moreover, protein degradation was faster in RF compared to PF which is shown by the amount of carbonyl content as well as western blot.”
Point 6: Discussion:
On line 529, the authors state “The difference might be related to the different feed ingredients.” This is correct, and is the reason why such data should have been collected.
Response 6: We agree to the reviewers comment. We would like to refer you to the point 2 tables. However, the results confirm that in general natural feed gives a higher nutritional value. Moreover, fish reared in the pond system just used natural feeds. Due to the big variety of feeds in pond it would be difficult to collect all the data. Furthermore, we would like to consider the effect of two rearing systems on the nutritional quality of perch fillets. Therefore, investigation on fatty acid profile provided good information about it. Additionally, it might be interesting for fish producers to know, which kind of rearing systems give a better fillet quality.
Point 7: On Lines 567-569, the authors write “Higher liquid loss and firmness in the RF compared to the PF might be related to higher water temperatures and lower swimming activity in the RAS system. However, the authors should have used Tukey’ Studentized Range Test to statistically analyze significant differences between the means. If they had done this on this analyses and other analyses, it would potentially have shown no significant differences. The Tukey's honestly significant difference test (Tukey's HSD) is used to test differences among sample means for significance. The Tukey's HSD tests all pairwise differences while controlling the probability of making one or more Type I errors.
Response 7: We appreciate your comments about this issue. We considered the statistical analysis for firmness and liquid loss and corrected in lines 464-465 and 500-501.
“RF showed higher hardness after 8 months frozen storage compared to PF”.
“We did not observe any significant differences between RF and PF after 12 days.”.
Point 8: Lines 614-616, the authors write “formation of ice crystals followed by cell disruption can be considered as a key role in the firmness deterioration rather than protein denaturation during the first months of frozen storage….” This is correct, the cause of the large ice crystals and cell disruption maybe due to the fact that these fillets were slow frozen rather than quick (i.e. blast) freezing (See Above). Blast freezing will result in smaller ice crystal formation and a better-quality product especially if afterwards these fillets are properly packaged and stored at -18C.
Response 8: Thank you for your constructive comment. The statement about the role of ice crystal formation and protein denaturation on firmness deterioration is our propose for the current study. In this study, all the process were similar for fillets from both rearing systems. Therefore, the influence of ice crystal formation would expect to have similar impact on both fillets.
Reviewer 2 Report
There are few comments and suggestions for consideration:
- All the scientific (Latin) names should be provided in italic. The comment applies to the entire text of the publication, including the title.
- Materials and Methods
- Authors performed numerous analyses and it is very difficult to track which of them were performed on chilled (0-142h) and which on frozen (0-8 months) raw material. It would be a great help to include such information for each description (subchapter) of method;
- Line 124-125 it was shown that water holding capacity (WHC) was assessed for the fillets, however in the section 2.7. description of the liquid loss was specified. WHC it's not the same as liquid loss;
- Section 2.5 Biogenic Amines – There is no reference for the equation to calculate QI and BAI. Please check equations. According to KAROVIÄŒOVÁ J., KOHAJDOVÁ Z. (Biogenic amines in foods. Chem. Pap. 2005. 59(1), 70-79) BAI = (mg kg−1 histamine + mg kg−1 putrescine + mg kg−1 cadaverine)/(1 + mg kg−1 spermine + mg kg−1 spermidine);
- Section 2.6 - firmness or more correct form i.e. hardness? There is no information whether the measurement was made on raw meat or it was heat treated in advance.
- Results
- Please unify the way of marking the differences between the samples, e.g. in the table 1 the lack of difference between is marked using “a” and “a” (n-3) or the marks are missing (C16:0);
- Table 1 - results for AI and TI are missing (the equations are provided in the Material and Methods). It can hardly be explained by the fact that there are no differences between the breeding systems because also there are no differences in the case of n-3 and n-6, which are shown in the table;
- Figure 1 – lack of „pond” in the legend;
- Table 2 – lack of results for BAI and QI, and no indication of differences for all parameters between storage times;
- Section 3.4., Authors use the different units for firmness - Newton (N) is used in description, but in methodology and in figures gram (g) is used Additionally, „The initial firmness value were 1533 (N) and 1239 (N) and decreased to 790 and 518 during frozen storage and declined to 1185 and 981 during refrigerated storage in the RF and PF, respectively”. 518-1533N in general is too high! In case of meat of farm and wild animals this value is much lower;
- Line 394 – please correct with „PF”
- Figure 4B – please check the significance of the differences on the 8th and 12th days of storage. In my opinion, the RF and PF trials did not differ significantly at these time points. Please correct the figure and description in the section 3.5;
- In the title of the Table 3b please include „RAS”;
- Figure 5B has low quality and difficult to read;
- Section 3.9.1. no information available on the main components of PC 1 and PC2 (type of storage, culture system?).
- Discussion
- Line 528 – please replace „good” with „better”
- Lack of discussion on the nutritional value of the fat in the muscle (AI and TI indexes);
- Line 537-539 - The difference in the pH of the RF and PR perch meat results rather from the different swimming activity of the fish, which is associated with different muscle metabolism and, consequently, different glycolysis rates. The change in the pH of meat during storage is primarily determined by the formation of lactic acid, and not by the „microbial activities and accumulation of ammonia”;
- Lack of information on BAI, which is used to determine quality decline in fish and fish products. A BAI value exceeding 10 is regarded as representing some kind of loss in quality;
- Line 570 – “…a more, darker…”?
Author Response
Point 1: All the scientific (Latin) names should be provided in italic. The comment applies to the entire text of the publication, including the title.
Response 1: Please find the italic form of all scientific names in lines 2, 53, 54, 89, 125, 127, 698.
Point 2: Materials and Methods
Authors performed numerous analyses and it is very difficult to track which of them were performed on chilled (0-142h) and which on frozen (0-8 months) raw material. It would be a great help to include such information for each description (subchapter) of method;
Response 2: Yes, we agree. We included subchapter in lines 134 and 152 to describe the methodology clearer.
2.1.1. Sample preparation for textural and chemical analysis
2.1.2. Sample preparation for rigor index
Point 3: Line 124-125 it was shown that water holding capacity (WHC) was assessed for the fillets, however in the section 2.7. description of the liquid loss was specified. WHC it's not the same as liquid loss;
Response 3: Yes, we corrected to liquid loss in line 142.
Point 4: Section 2.5 Biogenic Amines – There is no reference for the equation to calculate QI and BAI. Please check equations. According to KAROVIÄŒOVÁ J., KOHAJDOVÁ Z. (Biogenic amines in foods. Chem. Pap. 2005. 59(1), 70-79) BAI = (mg kg−1 histamine + mg kg−1 putrescine + mg kg−1 cadaverine)/(1 + mg kg−1 spermine + mg kg−1 spermidine);
Response 4: We agree. We included the reference and equation in lines 217-218.
Point 5: Section 2.6 - firmness or more correct form i.e. hardness? There is no information whether the measurement was made on raw meat or it was heat treated in advance.
Response 5: We change it accordingly in section 2.6. Regarding the fillet situation we included this information in lines 145-148.
“For the textural analysis we considered fresh fillet (immediately after slaughtering) as time 0. In case of, frozen samples all raw fillets were defrosted (kept at +4 °C over-night) in advance before, texture was measured.”
Point 6: Please unify the way of marking the differences between the samples, e.g. in the table 1 the lack of difference between is marked using “a” and “a” (n-3) or the marks are missing (C16:0);
Response 6: We corrected C16:0 in table 1. In case of C16:0 there is no difference between both rearing systems. Therefore, we marked both “a” and “a” in the table1.
Point 7: Table 1 - results for AI and TI are missing (the equations are provided in the Material and Methods). It can hardly be explained by the fact that there are no differences between the breeding systems because also there are no differences in the case of n-3 and n-6, which are shown in the table;
Response 7: We included AI and TI information in the table 1 accordingly. In this section we mentioned that there is no differ according to the statistical analysis.
Point 8: Figure 1 – lack of „pond” in the legend;
Response 8: Please find `pond´ included in the legends at the top of the figure.
Point 9: Table 2 – lack of results for BAI and QI, and no indication of differences for all parameters between storage times
Response 9: Yes, that is right. We considered BAI in our manuscript and ignored QI. Please find BAI information in table 2. Also, we included the statistical information according to the storage time in table 2.
Point 10: Section 3.4., Authors use the different units for firmness - Newton (N) is used in description, but in methodology and in figures gram (g) is used Additionally, „The initial firmness value were 1533 (N) and 1239 (N) and decreased to 790 and 518 during frozen storage and declined to 1185 and 981 during refrigerated storage in the RF and PF, respectively”. 518-1533N in general is too high! In case of meat of farm and wild animals this value is much lower;
Response 10: That is right. We corrected to gram (g) in line 462.
Point 11: Line 394 – please correct with „PF”
Response 11: We believe that is in the correct form.
Point 12: Figure 4B – please check the significance of the differences on the 8th and 12th days of storage. In my opinion, the RF and PF trials did not differ significantly at these time points. Please correct the figure and description in the section 3.5;
Response 12: Yes, that is right. We corrected in the figure 4b as well as in lines 500-501.
Point 13: In the title of the Table 3b please include „RAS”;
Response 13: We corrected in line 512.
Point 14: Figure 5B has low quality and difficult to read;
Response 14: Please accept our apology for the inconvenience. We put a better quality image for western blot analysis.
Point 15: Section 3.9.1. no information available on the main components of PC 1 and PC2 (type of storage, culture system?).
Response 15: We included this information in line 615 and 619.
Point 16: Line 528 – please replace „good” with „better”
Response 16: We changed it in line 650.
Point 17: Line 528 – Lack of discussion on the nutritional value of the fat in the muscle (AI and TI indexes);
Response 17: We included in lines 641-642.
“Lipid quality index AI and TI were similar (below 1) in RF and PF which indicate very good nutritional value for human health”.
Point 18: Line 537-539 - The difference in the pH of the RF and PR perch meat results rather from the different swimming activity of the fish, which is associated with different muscle metabolism and, consequently, different glycolysis rates. The change in the pH of meat during storage is primarily determined by the formation of lactic acid, and not by the „microbial activities and accumulation of ammonia”;
Response 18: We agree and mentioned that the formation of lactic acid is the main reason for decreasing pH. That is right that different pH between RF and PF is related to muscle metabolism and different glycolysis rates. However, we also explained the proposed reason that pH increased during the storage time due to the microbial activity.
“Formation of lactic acid might be the main reason for reduction of pH in both fillet until 24 hours storage. On the other hand, significantly higher pH after 72h storage in PF than RF can be related to microbial activities and accumulation of ammonia. Fur-therefore, upper situation of pond in the PCA plot compared to RAS confirmed higher pH in the PF rather than RF over the storage time.”
Point 19: Lack of information on BAI, which is used to determine quality decline in fish and fish products. A BAI value exceeding 10 is regarded as representing some kind of loss in quality;
Response 19: We agree. We included this information in lines 102-104 and 688-695.
Point 20: Line 570 – “…a more, darker…”?
Response 20: We corrected in line 702.
Reviewer 3 Report
Reviewer's Comment:
The manuscript describes the chemical change in Eurasian perch fillets deriving from two different rearing system during their conservation at frozen and refrigerate condition.
Title
Lines 2-3: the scientific name of fish must be in italic (here and in all the manuscript line 39, 40, ect…). The title must be improved, only chemical analysis have been studied.
Abstract
Lines 22-35: the abstract must be rewritten, is not clear. The conclusions are not present.
Keywords
Line 36: remove the “)”
Introduction
Line 39,40,72, ect.. : the scientific name of fish must be in italic (here and in all the manuscript line 39, 40, ect…)
Line 42: The abbreviation “RAS” cannot be presented for the first time in this way…it must be written in extenso.
Lines 43-45: insert at least one reference.
Lines 47-49: insert at least one reference.
Lines 60-62: insert at least one reference.
Lines 63-65: not clear sentence, these 3 mechanisms are related to fishes under refrigerate condition or frozen?
Lines 66-68: this sentence is not correct… fished are not reach in fat and the reduction of their shelf-life is not related to the rancidity but to the bacteria multiplication and the formation of biogenic ammines. This is the main concern: there is no introductory reference to the physical and chemical repercussions of microbial spoilage. This lack is found throughout the work, limiting its significance starting with the title. We cannot speak of quality without any reference to bacterial activity, but only of some qualitative aspect.
Materials and Methods
Lines 106-108: the scientific name of fish must be in italic
Lines 115-117: several information are not reported: in the week before slaughter, were the fish fed or not? If so, how?
The biometric data of the fish (weight, length) are absolutely not reported. Differences in size can greatly influence some of the aspects considered. The method of slaughter (percussion) is not sufficiently clarified.
Consideration and conclusion
In the considerations and conclusions it is not clear in the comparison between the two types of breeding how much I may have influenced a different type of diet. Also the influence of bacterial activity on some of the chemical indicators tested, such as biogenic amines and even the firmness, is not discussed.
Author Response
Point 1: Lines 2-3: the scientific name of fish must be in italic (here and in all the manuscript line 39, 40, ect…). The title must be improved, only chemical analysis have been studied.
Response 1: Please find the italic form of all scientific names in lines 2, 53, 54, 89, 125, 127, 698. Regarding the title we believe that quality changes consider all aspects of our study such as chemical (oxidation and biogenic amines development) as well as non-chemical (rigor mortis and textural parameters).
Point 2: Abstract
Lines 22-35: the abstract must be rewritten, is not clear. The conclusions are not present.
Response 2: We rewrote abstract accordingly. Please find in lines 24-36.
“The current knowledge on how different Eurasian perch rearing systems impact the final fillet quality is scant. Therefore, two domestic storage conditions were investigated; 10 months frozen (-20 °C) and 12 days refrigerated (+4 °C) storage in order to determine i) how the choice of rearing system affects fillets quality during different processing conditions and ii) if oxidative changes and other quality parameters were interactive. For the proposed idea proteome analysis, oxidative changes and some quality parameters were considered in this study. SDS-PAGE indicated higher loss of protein in frozen fillets from pond (PF) than the fillets from RAS (RF). Western blot showed higher protein carbonyls level in RF compared to PF, which was confirmed by total protein carbonyls during frozen storage. PF indicated less liquid loss, hardness and oxidation progress than RF in both storage conditions. The biogenic amines index (BAI) in the fillets from either origin showed ac-ceptable levels during storage at +4 °C. Furthermore, n-3/n-6 ratio was similar for both fillets. The deterioration of fillets during frozen storage were mainly caused by formation of ice crystal followed by protein oxidation while protein oxidation was the main concern during refrigerated storage confirmed by PCA analysis”.
Point 3: Keywords
Line 36: remove the “)”
Response 3: We corrected in line 52.
Point 4: Introduction
Line 39,40,72, ect.. : the scientific name of fish must be in italic (here and in all the manuscript line 39, 40, ect…)
Response 4: That is right. We corrected in lines 2, 53, 54, 89, 125, 127, 698.
Point 5: Line 42: The abbreviation “RAS” cannot be presented for the first time in this way…it must be written in extenso.
Response 5: We included the full name in line 56.
Point 6: Lines 43-45: insert at least one reference.
Response 6: We included in line 59.
Point 7: 47-49: insert at least one reference.
Response 7: We included in line 64.
Point 8: Lines 60-62: insert at least one reference.
Response 8: We included in line 72.
Point 9: Lines 63-65: not clear sentence, these 3 mechanisms are related to fishes under refrigerate condition or frozen?
Response 9: That is right. We mentioned in lines 79-81.
“It has been proposed that fish fillet softening occurs due to three mechanisms, namely, ice crystal formation during frozen storage [7], endogenous enzyme activity that led to protein degradation during refrigerated storage [8], and the progress of li-pid and protein oxidation and reaction product of these in both mentioned conditions”.
Point 10: Lines 66-68: this sentence is not correct… fished are not reach in fat and the reduction of their shelf-life is not related to the rancidity but to the bacteria multiplication and the formation of biogenic ammines. This is the main concern: there is no introductory reference to the physical and chemical repercussions of microbial spoilage. This lack is found throughout the work, limiting its significance starting with the title. We cannot speak of quality without any reference to bacterial activity, but only of some qualitative aspect.
Response 10: We agree. We corrected in lines 83 and 102-104.
“Rancidity of lipids leads to decrease the acceptability of fish fillet and make fillet prone to oxidation, especially the ones containing high amount of polyunsaturated fatty acids (PUFA)”.
“Biogenic Amines Index (BAI) is used as an indicator of freshness. In case of fish fillet 0 and 1 are reflect freshness, score between 1 and 10 are acceptable, and more than 10 demonstrates decomposition of fish fillet”.
Point 11: Materials and Methods
Lines 106-108: the scientific name of fish must be in italic
Response 11: That is right. We corrected in lines 125- 127.
Point 12: Lines 115-117: several information are not reported: in the week before slaughter, were the fish fed or not? If so, how?
Response 12: That is right. We included the information in line 135-136.
“Fish from both rearing systems were kept 7 days in flow-through tanks without feeding before slaughtering”.
Point 13: The biometric data of the fish (weight, length) are absolutely not reported. Differences in size can greatly influence some of the aspects considered. The method of slaughter (percussion) is not sufficiently clarified.
Response 13: That is right. We included biometric data in lines 119-122. We would not expect to observe any impact on our results due to the small different size.
“Eighty-eight Eurasian perch from two different rearing systems (traditional pond culture and RAS) (average 183 g and 211 g for 30 fish in weight and average 23.46 and 24.37 cm in length for RAS and pond fish respectively) with the pond age 2+ and RAS age 1+, were used in the present study”.
Fish were killed by blows to the head, bled and gutted consequently (please find this information in lines 136-139).
“Fish (both from RAS and pond) were kept at the faculty in tanks at the same temperature (20 °C) for one week before being slaughtered (fish were killed by blows to the head, bled and gutted, skinned, washed consequently and filleted on the faculty premises by one trained person”.
Point 14: Consideration and conclusion
In the considerations and conclusions it is not clear in the comparison between the two types of breeding how much I may have influenced a different type of diet. Also the influence of bacterial activity on some of the chemical indicators tested, such as biogenic amines and even the firmness, is not discussed.
Response 14: We appreciate your comments about this issue. We included some nutritional comparison between RF and PF in lines 774-778.
“Considering TI, AI and n-3/n-6 ratio, indicated almost same nutritional quality in both fillets. Additionally, we proposed that consumption of different diet influenced on colour as well as other quality parameters in current study. Therefore, concerning the final quality of the fillets from both rearing systems, we observed a better oxidation stability and lower BAI in the fillets from the pond system”.
Round 2
Reviewer 1 Report
The authors have addressed my main concerns, but there is one topic that still needs clarification. The authors state in the Abstract that freezing is the main reason for fillet deterioration. This statement needs clarification, since slow freezing of fillets will result in large ice crystal formation which will result in fillet deterioration during frozen storage. To prevent this deterioration, quick freezing, (i.e. blast freezing) and proper frozen storage conditions, will result in smaller ice crystal formation in the fillets, and the fillets will maintain better quality during frozen storage. Thus, it is improper freezing procedures that may have caused deterioration in the fillets. When done properly (i.e., quick freezing, proper packaging and stable frozen storage conditions) product quality will be maintained during frozen storage. In addition, the scientific soundness and significance of content would be high if the authors addressed this issue.
Author Response
Point 1: Comments and Suggestions for Authors
The authors have addressed my main concerns, but there is one topic that still needs clarification. The authors state in the Abstract that freezing is the main reason for fillet deterioration. This statement needs clarification, since slow freezing of fillets will result in large ice crystal formation which will result in fillet deterioration during frozen storage. To prevent this deterioration, quick freezing, (i.e. blast freezing) and proper frozen storage conditions, will result in smaller ice crystal formation in the fillets, and the fillets will maintain better quality during frozen storage. Thus, it is improper freezing procedures that may have caused deterioration in the fillets. When done properly (i.e., quick freezing, proper packaging and stable frozen storage conditions) product quality will be maintained during frozen storage. In addition, the scientific soundness and significance of content would be high if the authors addressed this issue.

Response 1: That is right. Please find in the lines 61-64 and 713-715.
Reviewer 3 Report
The authors have improved sufficently the paper.
Author Response
Thank you for considering our manuscript. There is no more comment to answer.